# SoLoPO: Unlocking Long-Context Capabilities in LLMs via Short-to-Long Preference Optimization

**Huashan Sun**[1][*]    **Shengyi Liao**[1][*]    **Yansen Han**    **Yu Bai**    **Yang Gao**[†]
**Cheng Fu**[1]    **Weizhou Shen**[1]    **Fanqi Wan**[1]    **Ming Yan**[1][†]    **Ji Zhang**[1]    **Fei Huang**[1]
[1] Tongyi Lab, Alibaba Group
{liaoshengyi.lsy,ym119608}@alibaba-inc.com
hanyansen@gmail.com  {hssun,yubai,gyang}@bit.edu.cn

## Abstract

Despite advances in pretraining with extended context sizes, large language models (LLMs) still face challenges in effectively utilizing real-world long-context information, primarily due to insufficient long-context alignment caused by data quality issues, training inefficiencies, and the lack of well-designed optimization objectives. To address these limitations, we propose a framework named **Sh**ort-to-**Lo**ng **P**reference **O**ptimization (**SoLoPO**), decoupling long-context preference optimization (PO) into two components: short-context PO and short-to-long reward alignment (SoLo-RA), supported by both theoretical and empirical evidence. Specifically, short-context PO leverages preference pairs sampled from short contexts to enhance the model's contextual knowledge utilization ability. Meanwhile, SoLo-RA explicitly encourages reward score consistency for the responses when conditioned on both short and long contexts that contain identical task-relevant information. This facilitates transferring the model's ability to handle short contexts into long-context scenarios. SoLoPO is compatible with mainstream preference optimization algorithms, while substantially improving the efficiency of data construction and training processes. Experimental results show that SoLoPO enhances all these algorithms with respect to stronger length and domain generalization abilities across various long-context benchmarks, while achieving notable improvements in both computational and memory efficiency[1].

## 1 Introduction

Long-text modeling is a cornerstone capability of large language models (LLMs) [76, 17, 77, 38]. While the input context size of LLMs has increased dramatically [44, 15], studies show that they can effectively utilize only 10–20% of this capacity primarily due to insufficient long-context alignment [36, 31, 65, 11], leaving their potential in long-context scenarios largely untapped.

To address this issue, data augmentation methods [43, 80, 4, 92, 75] leverage advanced LLMs to generate long-dependency instruction-following data for supervised fine-tuning (SFT) and preference optimization (PO). However, as text length increases, these methods suffer from declining reliability and efficiency. Moreover, directly applying short-context training strategies to long-context scenarios may overlook the inherent discrepancies between the two settings, yielding suboptimal performance [15, 37]. A different approach improves long-text alignment via training objective optimization. Fang et al. [18] propose LongCE, which identifies tokens critical for long-text modeling and assign them higher loss weights during SFT. However, this approach incurs extra computation due to multiple forward passes to identify salient tokens. LongPO [11] leverages responses generated with short contexts as positive examples in long-context direct preference optimization (DPO) [54]. Additionally, a short-to-long constraint is introduced, which optimizes the DPO objective by replacing

---

[*] Equal contribution.

[†] Corresponding authors

[1]Code and data resources are available at https://github.com/shs910/SoLoPO

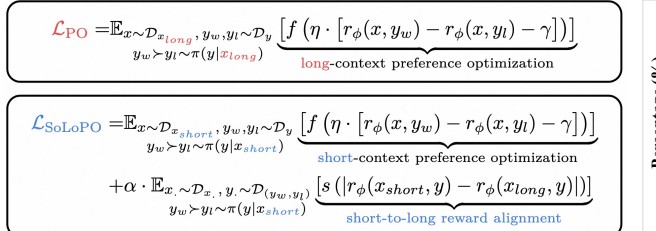 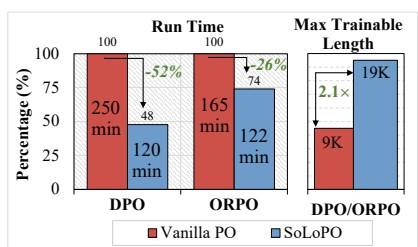

(a) Comparison of objectives between original PO and SoLoPO. $x_{long}$ denotes the original long-context input, and $x_{short}$ denotes the compressed short-context input preserving key task information.

(b) Efficiency of Vanilla PO vs. SoLoPO. SoLoPO greatly cuts the original PO run time and doubles the max trainable length.

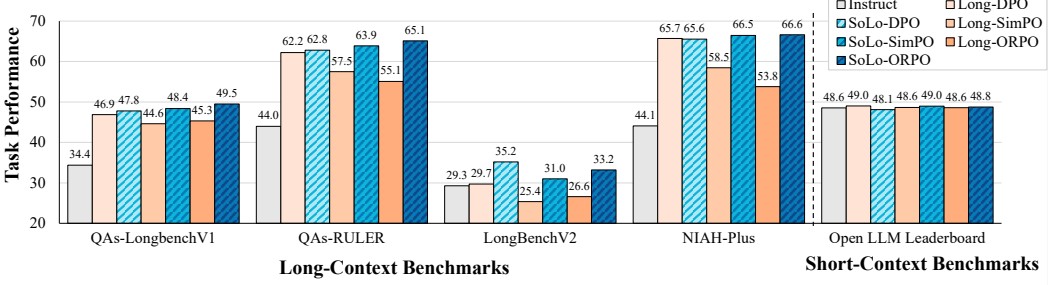

(c) Performance comparison of Qwen2.5-7B-Instruct [76] trained with original PO versus SoLoPO across various long-context and short-context benchmarks. Long-PO refers to original preference optimization on preference pairs sampled from long texts, while SoLo-PO denotes preference optimization within our SoLoPO framework using preference data derived from compressed short texts.

Figure 1: Original PO vs. SoLoPO. **(a)** SoLoPO decouples long-context PO into two components: short-context PO and short-to-long reward alignment, reducing the complexity of preference data construction and minimizing long-text processing during training. **(b)** Under identical configurations, SoLoPO exhibits superior training efficiency compared to vanilla methods. **(c)** SoLoPO outperforms the original PO across various long-context benchmarks while maintaining short-context ability.

$\pi_{ref}(y \mid x_{long})$ with $\pi_{ref}(y \mid x_{short})$ to mitigate performance degradation on short-context tasks. However, LongPO is not generalizable to other PO algorithms. Accordingly, long-context alignment poses three primary challenges: (1) difficulties in data construction, (2) inefficient training procedures, and (3) the absence of a suitable optimization objective.

In this paper, we introduce **Sh**ort-to-**Lo**ng **P**reference **O**ptimization (SoLoPO), a general, simple yet effective framework to transfer the powerful understanding ability of short contexts (hereafter termed short-context capability) of LLMs to long-context scenarios. As illustrated in Figure 1a, we first theoretically demonstrate that long-context PO [63] can be decoupled into two components: short-context PO and short-to-long reward alignment (SoLo-RA). Intuitively, SoLoPO enhances the model's contextual knowledge utilization ability via short-context PO, while SoLo-RA explicitly encourages the model to align reward scores between outputs conditioned on long and short contexts containing identical task-relevant information, thereby improving its long-context ability. SoLoPO offers three key advantages over existing methods: (1) sampling preference pairs from compressed shortened long contexts improves data quality and construction efficiency; (2) applying SoLo-RA only to the chosen responses reduces the burden of long-text processing during training, leading to more efficient optimization; (3) the optimization objective accounts for connections between short and long contexts, better favoring long-context alignment.

We apply SoLoPO to different PO algorithms, including DPO, SimPO [48], and ORPO [30]. As shown in Figure 1c, benefiting from SoLoPO, Qwen2.5-7B-Instruct trained solely on MuSiQue [64] achieves better performance across various domains and lengths on QA tasks from LongBenchV1 [5] and RULER [31], demonstrating stronger generalization than models trained with vanilla methods. Results on LongBenchV2 [6] and the Open-LLM-Leaderboard [19] further indicate that SoLoPO exhibits promising potential in handling contexts beyond pre-training window length, without com-

promising short-context capabilities. Further analysis on NIAH-Plus [84] indicates that SoLoPO 's decoupled approach with explicit SoLo-RA notably improves the contextual knowledge localization capability of LLMs. Moreover, SoLoPO significantly enhances efficiency, enabling $2.1\times$ longer trainable length under ZeRO stage 3 [55] with offloading while cutting run time by 52% and 26% for DPO and ORPO at $9K$ length, respectively (Figure 1b).

Our main contributions can be summarized as follows:

- We theoretically show that long-context PO can be decomposed into short-context PO and short-to-long reward alignment, providing new insights for long-context alignment.
- We propose SoLoPO, a general framework for long-context PO, which transfers the model's short-context ability to long-text scenarios while significantly improving training efficiency.
- We integrate mainstream preference optimization algorithms into SoLoPO and empirically demonstrate LLMs can have much better performance within this framework.

## 2 SoLoPO: Short-to-Long Preference Optimization

In this section, we first introduce the background of preference optimization (PO), including DPO and the unified framework, generalized preference optimization (GPO) [62] (§ 2.1). Then, by theoretically analyzing long-context preference modeling, we show that long-context PO can be decoupled into short-context PO and short-to-long reward alignment (§ 2.2). Based on this insight, we propose SoLoPO and apply it to various preference optimization algorithms (§ 2.3).

### 2.1 Preliminaries

**Reinforcement learning from human feedback (RHLF).** RHLF [49] aligns LLMs with human preferences through a two-stage process, further enhancing the model's capabilities. This involves training a reward model $r_\phi$ that captures human preferences, followed by regularized policy optimization to align the LLM with the learned reward model, more formally as below

$$\max_{\pi_\theta} \mathbb{E}_{x\sim\mathcal{D},y\sim\pi_\theta(y|x)} \left[ r_\phi(x,y) \right] - \beta\mathbb{D}_{KL} \left[ \pi_\theta(y|x)||\pi_{ref}(y|x) \right], \quad (1)$$

where $\pi_\theta$ is the policy model, $\pi_{ref}$ is the reference policy, typically initialized with the SFT model.

**Preference optimization (PO).** Without explicit reward modeling, DPO [54] reparameterizes the reward function using the optimal policy model and directly models the preference distribution by incorporating the Bradley-Terry ranking loss [10], enabling a single-stage preference alignment:

$$\mathcal{L}_{DPO}(\pi_\theta;\pi_{ref}) = -\mathbb{E}_{(x,y_w,y_l)\sim\mathcal{D}} \left[ \log\sigma \left( \beta\log\frac{\pi_\theta(y_w|x)}{\pi_{ref}(y_w|x)} - \beta\log\frac{\pi_\theta(y_l|x)}{\pi_{ref}(y_l|x)} \right) \right], \quad (2)$$

here, $(y_w, y_l)$ is a preference pair. Furthermore, Tang et al. [62] propose GPO, a unified framework for preference optimization, which allows us to parameterize the optimization objective using a convex function $f(\cdot)$ and hyperparameters $\eta$ and $\gamma$:

$$\mathcal{L}(r_\phi, \mathcal{D}) = \mathbb{E}_{(x,y_w,y_l)\sim\mathcal{D}} \left[ f\left( \eta\cdot(r_\phi(x,y_w) - r_\phi(x,y_l) - \gamma) \right) \right]. \quad (3)$$

### 2.2 Theoretical Analysis of Long-Context Preference Modeling

Recall that a key challenge in long-text alignment lies in the inefficiency of data construction and training. Thus, *can we represent long-context PO via short-context PO, thereby making data collection and training more tractable?* We analyze the upper bound of general long-context PO loss, demonstrating the viability of this approach based on the redundancy hypothesis.

**Redundancy hypothesis and compression rate.** Redundancy, pervasive in human language [69, 59], while potentially aiding human comprehension, may adversely affect LLMs [41, 50]. Particularly in task-aware scenarios [32, 42, 74], for a long context $c_{long}$ and a task instruction $I$, the model only needs to focus on relevant key content $c_{rel}$ while ignoring irrelevant content $c_{irr}$. Therefore, we can use *compression rate* [2] denoted as $\rho$, as a unified lens to observe long-context tasks, which refers to the information ratio between $c_{rel}$ and $c_{long}$. Most long-context tasks, such as question answering and information extraction, require only task-relevant excerpts from the source text [3, 32], yielding $\rho < 100\%$. As a special case, long-context translation [52, 28] has a compression rate $\rho = 100\%$.

**Problem setting.** Regarding long-context scenarios, we use $x_{long} := [c_{long}; I]$ to represent the input comprising the original long context $c_{long}$ and task instruction $I$. Based on the redundancy hypothesis, $c_{long}$ can be compressed, given the task instruction $I$, into a context $c_{rel}$ that preserves all task-critical information. We denote by $x_{short} := x_{rel} := [c_{rel}; I]$ the concatenation of $c_{rel}$ and $I$. For tasks with $\rho < 100\%$, $x_{short}$ is typically shorter than $x_{long}$; for $\rho = 100\%$, they are identical. Given a preference dataset $\mathcal{D}_{(x_{long}, y_w, y_l)}$, the objective of long-context PO is to model the preference relation $p(y_w \succ y_l \mid x_{long})$ by minimizing the preference modeling loss, as defined in Eq. (3).

**The upper bound of long-context general preference modeling loss.** To simplify notation in the following analysis, we define the preference loss in Eq. (3) for any given tuple $(x_1, x_2, y_1, y_2)$ as:

$$l_{\eta,\gamma}(x_1, x_2; y_1, y_2) = f(\eta \cdot [r_\phi(x_1, y_1) - r_\phi(x_2, y_2) - \gamma]). \tag{4}$$

The expectation of Equation (4) can then be expressed as:

$$\mathcal{L}_{\eta,\gamma}(\mathcal{D}_{x_1}, \mathcal{D}_{x_2}; \mathcal{D}_{y_1}, \mathcal{D}_{y_2}) = \mathbb{E}_{x_1, x_2 \sim \mathcal{D}_{x_1}, D_{x_2}; y_1, y_2 \sim \mathcal{D}_{y_1}, \mathcal{D}_{y_2}}[l_{\eta,\gamma}(x_1, x_2; y_1, y_2)]. \tag{5}$$

**Assumption 1** (Discrimination of preference order). *Based on the redundancy hypothesis, $x_{long} := [(c_{rel}, c_{irr}); I]$ contains more task-irrelevant information compared to $x_{short} := [c_{rel}; I]$, which may hinder LLMs' task performance [41, 50]. Consequently, distinguishing the order between $y_w$ and $y_l$ given $x_{long}$ is more difficult than given $x_{short}$ (refer to Appendix I.7 for experimental evidence):*

$$p(y_w \succ y_l \mid x_{long}) \leq p(y_w \succ y_l \mid x_{short}) \tag{6}$$

*When $x_{short}$ and $x_{long}$ are identical, the equality holds, giving a compression rate $\rho$ of 100%.*

Building upon the previous preparations, we establish the theoretical upper bound for the optimization objective over long-context data in Theorem 1. This bound provides formal guarantees that optimizing on short-context data while maintaining robust long-context performance is theoretically feasible.

**Theorem 1** (Relation between long-context and short-context preference optimization losses). *Under assumption 1, suppose $f$ is a convex function and satisfies $f(x + \gamma) + f(-x + \gamma) \leq s(|x|)$ for some function $s(\cdot)$ and non-negative constant $\gamma, \eta$. Then the following inequality holds:*

$$\mathcal{L}_{\eta,\gamma}(x_{long}) \leq \frac{1}{3}[\mathcal{L}_{3\eta, \frac{\gamma}{3}}(x_{short}) + \mathbb{E}_{x. \sim \mathcal{D}_{x.}; y \sim \mathcal{D}_y} s(|3\eta \cdot (r_\phi(x_{short}, y) - r_\phi(x_{long}, y))|)] \tag{7}$$

*where $\mathcal{L}_{\eta,\gamma}(x_{text}) := \mathcal{L}_{\eta,\gamma}(\mathcal{D}_{x_{text}}, \mathcal{D}_{x_{text}}; \mathcal{D}_{y_w \succ y_l | x_{text}}, \mathcal{D}_{y_w \succ y_l | x_{text}})$*

The complete derivation of Theorem 1 is presented in Appendix I.2, where $s(\cdot)$ is introduced with the primary objective of providing a metric to quantify the distance between $r_\phi(x_{short}, y)$ and $r_\phi(x_{long}, y)$. Given that $s(\cdot)$ serves as an upper bound, a tighter instantiation is theoretically preferred; we provide empirical evidence for this claim in Appendix I.8. Additionally, an extension of Theorem 1 is provided in Appendix I.5, which potentially holds promise for wider applicability.

**Objective Function of Short-to-Long Preference Optimization (SoLoPO).** Based on the theorem 1, we can define the general formula of the SoLoPO loss function:

$$\mathcal{L}_{SoLoPO} = \mathbb{E}_{\substack{x \sim \mathcal{D}_{x_{short}}; y_w, y_l \sim \mathcal{D}_y \\ y_w \succ y_l \sim \pi_\theta(y|x_{short})}} \underbrace{\left[ f\left( 3\eta \cdot [r_\phi(x, y_w) - r_\phi(x, y_l) - \frac{\gamma}{3}] \right) \right]}_{\text{short-context preference optimization}} \tag{8}$$

$$+ \alpha \cdot \mathbb{E}_{\substack{x. \sim \mathcal{D}_{x.}; y. \sim \mathcal{D}_{(y_w, y_l)} \\ y_w \succ y_l \sim \pi_\theta(y|x_{short})}} \underbrace{[s(3\eta \cdot |r_\phi(x_{short}, y) - r_\phi(x_{long}, y)|)]}_{\text{short-to-long reward alignment}}. \tag{9}$$

Here, $\gamma$, $\eta$ and $f(\cdot)$ are specified by the original PO algorithm, and $s(\cdot)$ satisfies $f(x + \gamma) + f(-x + \gamma) \leq s(|x|)$. $\alpha$ is a hyperparameter balancing the two loss terms. Thus, we theoretically decouple long-context PO into short-context PO and short-to-long reward alignment. Specifically, we present detailed derivations of the SoLoPO objective from Theorem 1 for two common convergence functions, $f(x) = x^2$ and $f(x) = -\log \sigma(x)$, in Appendices I.3 and I.4, respectively. Table 16 lists further examples of $f(\cdot)$ and their associated $s(\cdot)$. Moreover, the analysis in Section 4.2 provides experimental evidence supporting the validity of this decoupling. When $\rho = 100\%$, $x_{long}$ and $x_{short}$ are identical, rendering SoLoPO equivalent in form to the original PO, with differences confined solely to $\eta$ and $\gamma$; for example, in long-context machine translation, the entire context is task-relevant.

**Short-to-long reward alignment (SoLo-RA).** As shown in Eq. (9), SoLo-RA implies that, under optimal conditions, the reward model $r_\phi$ should assign a consistent score to response $y$ when conditioned on either $x_{long}$ or $x_{short}$, as long as the input retains all task-relevant information $c_{rel}$.

**What does the SoLoPO learn?** Unlike short-context tasks such as mathematics [58, 39] that draw upon the LLMs' intrinsic reasoning ability, long-context tasks necessitate a two-step process: first, identifying critical information within the given context, and second, executing the task based on that located information [38]. Consequently, proficient in both contextual knowledge localization and contextual knowledge utilization or reasoning. Compared to vanilla PO algorithms, which lack explicit modeling of the former, SoLoPO's decoupled optimization process is better aligned with these requirements, potentially leading to superior performance due to its distinct focus on: (a more detailed discussion can be found in Appendix I.9)

- **Contextual knowledge localization.** SoLo-RA (Eq. (9)) forces the reward model to implicitly predict $\hat{x}_{short} \sim \hat{p}(x_{short}|x_{long})$, minimizing divergence between predicted $\hat{x}_{short}$ and actual $x_{short}$. In preference optimization, since the reward model is the policy model itself, this also improves the policy model's ability to identify task-relevant knowledge within long context.

- **Contextual knowledge reasoning.** Since $x_{short}$ contains all task-relevant information, short-context PO (Eq. (8)) enhances the model's reasoning ability over this contextual knowledge.

**Non-decoupled short-to-long alignment.** Based on the above discussion, another non-decoupled approach to short-to-long alignment involves directly applying preference pairs sampled from short texts for long-context PO or SFT, which we term *Expand-Long-PO* and *Expand-Long-SFT*, respectively. Experiments in Section 4.2 show that the decoupled approach yields superior performance.

## 2.3 Applications of Short-to-Long Preference Optimization

**Chosen-only short-to-long reward alignment (chosen-only SoLo-RA).** Considering that $y_l \sim \pi_\theta(y|x_{short})$ may not fully exploit task-relevant contextual information (*e.g.*, responses of "No answer"), performing SoLo-RA on $y_l$ might introduce negative effects on model learning. A supporting analysis of this issue is provided in Appendix I.10.2. Therefore, to further improve training efficiency, we only apply SoLo-RA on $y_w$. Experimental analysis in Section 4 demonstrates the effectiveness of this approach, which also reduces training resource consumption while maintaining training stability.

Table 1: Applications of SoLoPO to mainstream PO algorithms: DPO, SimPO, and ORPO. **1.** Only the chosen-only SoLo-RA is shown; the short-context PO formulation is identical to the original algorithms. **2.** For DPO, $\pi_{ref}$ is omitted since it does not involved in differentiation.

| Original Method | Reward | Chosen-only SoLo-RA |
|---|---|---|
| DPO [54] | $\beta \log \frac{\pi_r(y_w|x)}{\pi_{ref}(y_w|x)} + \beta \log Z(x)$ | $\|\beta \log \pi_\theta(y_w|x_{short}) - \beta \log \pi_\theta(y_w|x_{long})\|$ |
| SimPO [48] | $\frac{\beta}{|y_w|} \log \pi_\theta(y_w|x)$ | $\|\frac{\beta}{|y_w|} \log \pi_\theta(y_w|x_{short}) - \frac{\beta}{|y_w|} \log \pi_\theta(y_w|x_{long})\|$ |
| ORPO [30] | $\log \frac{\pi_\theta(y_w|x)}{1-\pi_\theta(y_w|x)}$ | $\|\log \frac{\pi_\theta(y_w|x_{short})}{1-\pi_\theta(y_w|x_{short})} - \log \frac{\pi_\theta(y_w|x_{long})}{1-\pi_\theta(y_w|x_{long})}\|$ |

SoLoPO can be applied to various PO algorithms, provided that the corresponding convergence function $f(\cdot)$ and upper bound function $s(\cdot)$ are specified (see Table 16). We apply SoLoPO to mainstream algorithms, including DPO, SimPO, and ORPO, with their corresponding optimization objectives shown in Table 1. For brevity, we only present the expressions for the chosen-only SoLo-RA, while the objective functions for short-context PO remain consistent with the original methods. For DPO, since $\pi_{ref}(y_w \mid x)$ is constant and not involved in differentiation, we only align $\pi_\theta(y_w \mid x)$. See Appendix I.12 for the complete derivation and expressions. Unless otherwise stated, SoLoPO refers to its chosen-only SoLo-RA variant in the remainder of this paper.

**How does SoLoPO improve data sampling and training efficiency?** As illustrated in Eq. (8), SoLoPO's preference sampling leverages $x_{short}$, which, due to its shorter length and lower processing complexity compared to $x_{long}$, enables faster and more effective sampling of high-quality preference pairs. Furthermore, by applying chosen-only SoLo-RA, we process $x_{long}$ once and $x_{short}$ twice per training step. This contrasts with vanilla PO needing two $x_{long}$ passes, where $x_{long}$ processing is

significantly more costly than $x_{short}$. Thus, SoLoPO substantially boosts training efficiency. As the lengths of $x_{short}$ and $x_{long}$ become more similar (higher $\rho$), the efficiency gain from SoLoPO diminishes. Further detailed analysis and potential optimization methods are discussed in Appendix H.5.

Table 2: Composition of different datasets and corresponding trained models. **1.** SoLo denotes short-to-long alignment, where preference pairs derived from short contexts are used for long-context alignment. **2.** "*" means the corresponding PO method used in SoLoPO. **3.** $D^{\text{SoLo}}$ is also utilized for training LongPO, which falls under non-decoupled Short-to-Long DPO in our framework.

| Method | Dataset | Trained Models |
|---|---|---|
| SFT | $D^{\text{sft}}_{\text{short}} = \{(q, x_{\text{short}}, y^{\text{short}}_w)\}$ | $M^{\text{SFT}}_{\text{short}}$ |
| | $D^{\text{sft}}_{\text{long}} = \{(q, x_{\text{long}}, y^{\text{long}}_w)\}$ | $M^{\text{SFT}}_{\text{long}}$ |
| PO | $D^{\text{po}}_{\text{short}} = \{(q, x_{\text{short}}, y^{\text{short}}_w, y^{\text{short}}_l)\}$ | $M^{\text{PO}}_{\text{short}}$ |
| | $D^{\text{po}}_{\text{long}} = \{(q, x_{\text{long}}, y^{\text{long}}_w, y^{\text{long}}_l)\}$ | $M^{\text{PO}}_{\text{long}}$ |
| SoLo | $D^{\text{sft}}_{\text{expand-long}} = \{(q, x_{\text{long}}, y^{\text{short}}_w)\}$ | $M^{\text{SFT}}_{\text{expand-long}}$ |
| | $D^{\text{po}}_{\text{expand-long}} = \{(q, x_{\text{long}}, y^{\text{short}}_w, y^{\text{short}}_l)\}$ | $M^{\text{PO}}_{\text{expand-long}}$ |
| | $D^{\text{SoLo}} = \{(q, x_{\text{short}}, x_{\text{long}}, y^{\text{short}}_w, y^{\text{short}}_l)\}$ | $M^{(*)}_{\text{SoLo}}$ |

## 3 EXPERIMENTAL SETUP

**Dataset Construction** We construct $x_{\text{short}}$ and $x_{\text{long}}$ from the MuSiQue [64] training set using the method in RULER [31]. Specifically, we form a long context by mixing relevant documents with random unrelated ones. On average, the short and long context contain 1.1K and 7.5K tokens, respectively. We use the original QA-pairs $(q, a)$ from Musique as the questions and ground truth answers. To obtain preference pairs, we perform sampling with a temperature of 0.85 using the instruction model. For each input $(x_{\text{short}}, q, a)$, we sample $N = 32$ Chain-of-Thought [67] outputs and then select the corresponding preference pairs $(y^{\text{short}}_w, y^{\text{short}}_l)$ using the sub-em metric. Ultimately, we synthesize 5,000 training samples $D = \{(x_{\text{long}}, x_{\text{short}}, q, a, y^{\text{short}}_w, y^{\text{short}}_l)\}$. Additionally, we also sample 5,000 real long-context preference pairs $(y^{\text{long}}_w, y^{\text{long}}_l)$ based on $x_{\text{long}}$. The composition of different datasets is shown in Table 2. Figure 5 shows the pipeline and more details are in Appendix D.

**Baselines and Models** As shown in Table 2, we compare SoLoPO with other approaches that perform SFT or original PO on different datasets. Additionally, we incorporates results from LongPO [11], which optimizes the reward of DPO based on short-to-long KL constraint, replacing $\pi_{ref}(y \mid x_{\text{long}})$ with $\pi_{ref}(y \mid x_{\text{short}})$. We also introduce results from Qwen2.5-Instruct-32B/72B and Llama3.1-Instruct-70B for comparative analysis. We use Qwen2.5-7B-Instruct [76] and Llama3.1-8B-Instruct [34] as the backbones for our experiments with per-training context window of $32K$ and $128K$, respectively (hereafter, Qwen2.5-Instruct and Llama3.1-Instruct are referred to as Qwen2.5 and Llama3.1 for brevity). Appendix H.6 presents experiments on Qwen2.5-Instruct-14B to evaluate the scalability of SoLoPO. More training details are provided in the Appendix E.1

**Evaluation benchmarks** To comprehensively analyze the effectiveness of SoLoPO, we conduct evaluations on both long-context and short-context benchmarks. The long-context benchmarks include: (1) Real-world QA tasks from LongBenchV1 [5], used to evaluate the generalization capability of different methods on multi-document and single-document question answering in real scenarios within a $32K$ context size. (2) Synthetic QA tasks based on RULER [31], used to evaluate the generalization capability of different methods across various context lengths ($4K/8K/16K/32K$). (3) We further leverage LongBenchV2 [6] to analyze SoLoPO's potential on longer and more diverse real-world long-context tasks, and employ NIAH-Plus [84] to examine different models' context knowledge utilization ability in Section 4.2. For short-context benchmarks, we use MMLU-Pro [66], MATH [27], GPQA [57], IFEval [87], and BBH [60], following Open LLM Leaderboard [19].

Following previous works [5, 89], we utilize F1-score and multiple-choice accuracy as evaluation metrics, based on task-specific formats. For a fair comparison, we select the best-performing checkpoint on LongBenchV1 within a single training epoch as the final model for cross-benchmark evaluation. See Appendix E.2 for a detailed description of evaluation settings and benchmarks.

# 4 EXPERIMENTAL RESULTS

In this section, we present our main experimental results highlighting the effectiveness of the SoLoPO framework across various benchmarks and PO methods (§ 4.1). Through comprehensive comparative analysis, we provide deeper insights into the key components of SoLoPO: decoupling and direct chosen-only short-to-long reward alignment and analyze the impact of the reward alignment coefficient $\alpha$ (§ 4.2). We then experimentally validate the efficiency advantage of SoLoPO (§ 4.3).

## 4.1 MAIN RESULTS

Table 3: Performance comparison on QA tasks from LongBenchV1 and RUELR. For LongPO, "pub" denotes the public checkpoint, while "reimp" indicates our implementation on $D^{\text{SoLo}}$. **Bold** and underlined indicate the best and the second-best performance, respectively.

| Model | QAs-LongBenchV1 | | | QAs-RULER | | | | |
|---|---|---|---|---|---|---|---|---|
| | S-Doc QA | M-Doc QA | Avg. | 4k | 8k | 16k | 32k | Avg. |
| Qwen2.5-72B-Instruct | 37.8 | 61.1 | 49.4 | 65.7 | 64.4 | 61.2 | 55.0 | 61.6 |
| Qwen2.5-32B-Instruct | 34.1 | 49.8 | 41.9 | 58.4 | 52.1 | 46.0 | 43.9 | 50.1 |
| Llama3.1-70B-Instruct | 28.5 | 64.1 | 46.3 | 72.1 | 68.8 | 52.4 | 23.2 | 54.1 |
| LongPO-Qwen2.5-7B(reimp) | 34.8 | 52.6 | 43.7 | 62.4 | 54.4 | 48.9 | 43.1 | 52.2 |
| LongPO-Qwen2.5-7B[11](pub) | 27.5 | 38.3 | 32.9 | 54.7 | 51.9 | 40.6 | 36.3 | 45.9 |
| **Qwen2.5-7B-Instruct** | | | | | | | | |
| Instruct | 29.4 | 39.4 | 34.4 | 53.9 | 50.1 | 37.6 | 34.6 | 44.0 |
| $M_{\text{short}}^{\text{SFT}}$ | 28.9 | 48.4 | 38.6 | 63.8 | 56.7 | 42.3 | 31.8 | 48.7 |
| $M_{\text{long}}^{\text{SFT}}$ | 34.8 | 55.8 | 45.3 | 65.9 | 61.4 | 58.4 | 52.4 | 59.5 |
| $M_{\text{short}}^{\text{DPO}}$ | 34.6 | 51.8 | 43.2 | 70.9 | 63.3 | 45.3 | 46.9 | 56.6 |
| $M_{\text{long}}^{\text{DPO}}$ | 35.7 | 58.2 | 46.9 | **71.0** | 64.2 | 60.6 | 53.2 | 62.2 |
| $M_{\text{SoLo}}^{\text{DPO}}$ | 38.0 | 57.6 | 47.8 | 66.4 | 64.5 | 62.7 | 57.7 | 62.8 |
| $M_{\text{short}}^{\text{SimPO}}$ | 34.7 | 53.6 | 44.1 | 70.8 | 62.5 | 42.1 | 48.8 | 56.0 |
| $M_{\text{long}}^{\text{SimPO}}$ | 34.2 | 54.9 | 44.6 | 69.8 | 64.1 | 49.1 | 47.2 | 57.5 |
| $M_{\text{SoLo}}^{\text{SimPO}}$ | **38.1** | 58.6 | 48.4 | 69.2 | 66.0 | 62.7 | **57.8** | 63.9 |
| $M_{\text{short}}^{\text{ORPO}}$ | 28.9 | 48.4 | 38.6 | 69.1 | 62.1 | 50.8 | 46.6 | 57.1 |
| $M_{\text{long}}^{\text{ORPO}}$ | 34.8 | 55.8 | 45.3 | 64.9 | 59.9 | 50.0 | 45.6 | 55.1 |
| $M_{\text{SoLo}}^{\text{ORPO}}$ | 37.6 | **61.4** | **49.5** | 70.8 | **68.3** | **64.0** | 57.3 | **65.1** |
| **Llama3.1-8B-Instruct** | | | | | | | | |
| Instruct | 30.3 | 49.3 | 39.8 | 58.3 | 49.2 | 42.9 | 35.6 | 46.5 |
| $M_{\text{short}}^{\text{SFT}}$ | 33.0 | 56.2 | 44.6 | **65.0** | 61.0 | 58.5 | 52.2 | 59.2 |
| $M_{\text{long}}^{\text{SFT}}$ | 35.0 | **57.3** | 46.1 | 63.7 | 59.0 | 57.5 | 53.5 | 58.4 |
| $M_{\text{short}}^{\text{ORPO}}$ | 33.1 | 55.1 | 44.1 | 64.0 | 59.3 | 59.7 | 50.4 | 58.4 |
| $M_{\text{long}}^{\text{ORPO}}$ | **35.4** | 55.4 | 45.4 | 63.2 | 60.1 | 58.9 | 53.2 | 58.9 |
| $M_{\text{SoLo}}^{\text{ORPO}}$ | 35.2 | **57.3** | **46.3** | 64.4 | 62.5 | 60.1 | 58.2 | 61.3 |

**SoLoPO effectively enhances long-context capabilities within pre-training windows.** As illustrated in Table 3, SoLoPO achieves substantial performance gains and strong generalization, outperforming the original PO algorithm in 28 out of 32 settings. Qwen2.5-7B, trained solely on the Musique [64] dataset, achieves a score comparable to Qwen2.5-72B on LongBenchV1. Compared to various original algorithms (DPO, SimPO, ORPO), it attains performance improvements of 0.9, 4.3, and 10.9 points, respectively. Furthermore, SoLoPO exhibits consistently superior performance across varying context lengths on RULER, with only a performance drop observed at the length of $4K$ for DPO and SimPO. Similarly, we conduct experiments with SoLo-ORPO, the best-performing approach, on Llama3.1-8B, and the results further validate our claims. Given that Llama3.1-8B has a context size of $128K$, a $32K$ test length may already lie within its effective range (*i.e.* 25% [31]), and our experimental data has a maximum length of $8K$; therefore, the gains are smaller compared to Qwen2.5-7B. Specifically, SoLo-ORPO attains performance gains of 4.2 vs. 0.9 on LongBenchV1 and 10.0 vs. 2.4 on RULER over Long-ORPO, for Qwen2.5-7B and Llama3.1-8B, respectively. See Appendix H.6 for scalability experiments on Qwen2.5-14B.

Table 4: Performance comparison of different models on LongBenchV2 and Open LLM Leaderboard. Red values indicate performance degradation on short-context tasks compared to the Instruct model.

| Model | LongBenchV2 | | | | | | Open LLM Leaderboard | | | | | |
|---|---|---|---|---|---|---|---|---|---|---|---|---|
| | Overall | Easy | Hard | <32k | 32k-128k | >128k | MMLU-Pro | IFEval | BBH | MATH | GPQA | Avg. |
| **Qwen2.5-7B-Instruct** | | | | | | | | | | | | |
| Instruct | 29.3(±0.7) | 30.9 | 28.3 | 36.9 | 24.6 | 26.1 | 44.63 | 74.22 | 55.25 | 36.86 | 31.88 | 48.56 |
| LongPO(pub) | 33.3(±0.5) | 35.0 | 32.0 | 40.5 | 30.0 | 27.8 | 44.69 | 76.49 | 53.94 | 32.32 | 31.87 | 47.86 |
| LongPO(reimp) | 29.6(±1.5) | 32.2 | 28.0 | 36.7 | 26.7 | 23.7 | 44.80 | 73.86 | 55.07 | 34.81 | 31.91 | 48.08 |
| $M^{SFT}_{short}$ | 30.8(±0.9) | 33.6 | 29.1 | 39.2 | 25.7 | 27.2 | 44.81 | 72.18 | 55.15 | 36.71 | 32.12 | 48.19 |
| $M^{SFT}_{long}$ | 30.0(±1.4) | 32.0 | 28.8 | 35.7 | 25.9 | 28.7 | 44.74 | 71.46 | 55.35 | 36.55 | 31.45 | 47.90 |
| $M^{ORPO}_{short}$ | 29.3(±1.1) | 34.4 | 26.2 | 35.0 | 25.3 | 27.8 | 44.78 | 74.70 | 55.26 | 38.21 | 30.95 | 48.78 |
| $M^{ORPO}_{long}$ | 26.6(±1.2) | 30.1 | 24.5 | 33.8 | 22.3 | 23.3 | 44.64 | 73.61 | 55.37 | 37.16 | 32.12 | 48.58 |
| $M^{ORPO}_{SoLo}$ | **33.2(±1.0)** | 36.3 | 31.2 | 39.7 | 28.8 | 30.9 | 44.83 | 75.18 | 55.23 | 37.16 | 31.46 | 48.77 |
| $M^{DPO}_{short}$ | 25.5(±1.0) | 27.2 | 24.5 | 29.7 | 23.5 | 22.8 | 44.91 | 75.06 | 54.99 | 39.12 | 31.21 | 49.06 |
| $M^{DPO}_{long}$ | 29.7(±0.7) | 34.3 | 26.9 | 35.9 | 25.6 | 27.6 | 44.91 | 75.30 | 55.06 | 37.99 | 31.80 | 49.01 |
| $M^{DPO}_{SoLo}$ | **35.2(±1.2)** | 37.5 | 33.8 | 39.3 | 31.8 | 35.0 | 44.66 | 73.98 | 54.78 | 35.57 | 31.63 | 48.12 |
| $M^{SimPO}_{short}$ | 24.6(±1.5) | 27.1 | 23.1 | 29.5 | 21.2 | 23.1 | 44.97 | 73.50 | 54.94 | 38.60 | 31.46 | 48.69 |
| $M^{SimPO}_{long}$ | 25.4(±0.3) | 25.4 | 25.4 | 33.0 | 20.2 | 23.3 | 44.74 | 73.50 | 55.29 | 37.61 | 32.05 | 48.64 |
| $M^{SimPO}_{SoLo}$ | **31.0(±1.3)** | 34.1 | 29.1 | 37.5 | 25.7 | 30.6 | 44.78 | 75.90 | 54.89 | 37.76 | 31.54 | 48.97 |
| **Llama3.1-8B-Instruct** | | | | | | | | | | | | |
| Instruct | 32.5(±1.0) | 35.5 | 30.7 | 40.7 | 27.7 | 28.5 | 37.79 | 62.23 | 50.98 | 15.25 | 31.71 | 39.59 |
| $M^{SFT}_{short}$ | 31.8(±1.7) | 34.5 | 30.0 | 38.3 | 28.6 | 27.0 | 36.37 | 60.31 | 50.23 | 17.90 | 31.79 | 39.32 |
| $M^{SFT}_{long}$ | 30.9(±1.3) | 36.8 | 27.3 | 36.0 | 29.6 | 24.8 | 37.04 | 60.67 | 49.64 | 16.16 | 32.38 | 39.17 |
| $M^{ORPO}_{short}$ | 33.7(±0.3) | 36.8 | 31.8 | 41.2 | 29.8 | 28.7 | 37.26 | 61.15 | 50.95 | 14.88 | 32.13 | 39.27 |
| $M^{ORPO}_{long}$ | 32.1(±1.1) | 34.5 | 30.6 | 39.7 | 27.8 | 28.0 | 37.43 | 60.43 | 50.86 | 15.86 | 31.63 | 39.24 |
| $M^{ORPO}_{SoLo}$ | **34.7(±0.9)** | 37.5 | 33.0 | 40.0 | 32.1 | 31.2 | 37.60 | 63.43 | 50.18 | 15.63 | 31.38 | 39.64 |

**SoLoPO shows better length generalization beyond the pre-training window.** We test Qwen2.5-7B (w/ YARN [51]) and Llama3.1-8B on LongBenchV2 with results presented in Table 4. For Qwen2.5-7B (w/ YARN), SoLoPO consistently outperforms the original PO algorithms across varying difficulty levels and context lengths. This highlights the superior generalization ability of models trained with SoLoPO, demonstrating its promise in handling longer and more diverse real-world long-context tasks. For Llama3.1-8B, SoLo-ORPO shows improved performance across all evaluation dimensions, except for a slight degradation on tasks with input length <32k words. While short-text training inherently aids length generalization [21, 90], SoLoPO's advantage likely stems from SoLo-RA, which explicitly enhances contextual knowledge localization, as discussed in Appendix I.9. Ablation experiments on NIAH-Plus in Appendix H.3 further support this claim. Additionally, for DPO, SoLo-DPO outperforms LongPO(pub), despite the latter is trained on a larger volume of longer data. This may stem from LongPO's direct assignment of $y \sim \pi(y|x_{long})$ as $y_l$ and $y \sim \pi(y|x_{short})$ as $y_w$ without ensuring $y_w \succ y_l$. In contrast, our method constructs preference data from ground truth, ensuring correctness and quality.

**SoLoPO maintains short-context performance.** As shown in Table 4 on the Open LLM Leaderboard, SoLoPO maintains short-context capabilities relative to the Instruct model, with only a slight decrease on DPO. This trade-off is acceptable as SoLoPO simultaneously enhances the model's long-context understanding ability and training efficiency. See Appendix I.11 for the supporting analysis of short-context stability in SoLoPO framework.

## 4.2 IN-DEPTH EXPLORATION OF SOLOPO

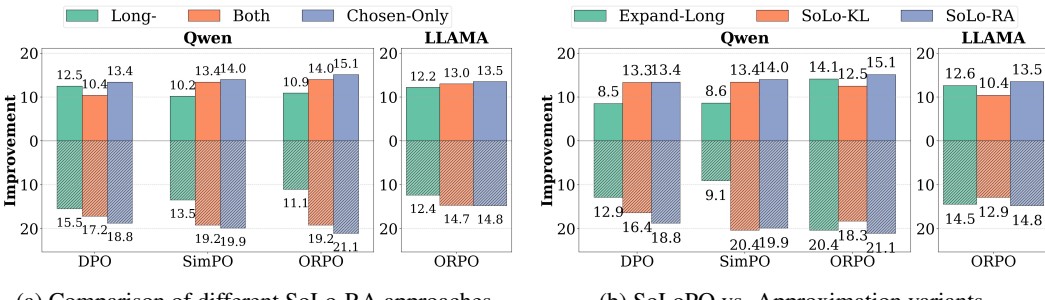

(a) Comparison of different SoLo-RA approaches.  (b) SoLoPO vs. Approximation variants.

Figure 2: Performance improvements of different short-to-long preference optimization frameworks based on various PO algorithms over Qwen2.5-7B on LongBenchV1 (top) and RULER (bottom).

**Empirical evidence for SoLoPO and superior performance of chosen-only SoLo-RA.** We investigate the impact of *chosen-only* SoLo-RA versus applying SoLo-RA jointly to both $y_w$ and $y_l$ (*both*), as defined in original SoLoPO (Eq.(9)). As shown in Figure 2a, SoLoPO with *both* SoLo-RA consistently outperforms Long-PO, except for a slight drop on DPO, supporting the validity of its decoupling strategy theoretically established in Section 2.2. Moreover, the *chosen-only* SoLo-RA surpasses the *both* approach across diverse algorithms and models. Appendix H.2 reveals that, compared with the *both* setting, the *chosen-only* version yields larger reward margins (Figure 10a) and assigns lower prediction probability to $y_l$ (Figure 10b). These results suggest that *chosen-only* SoLo-RA achieves stronger fitting capacity and more stable convergence, leading to better performance.

**Direct reward alignment matters.** To validate the effectiveness of SoLo-RA, we compare it with an approximation we termed short-to-long KL divergence (SoLo-KL):

$$\alpha \cdot \mathbb{E}_{x. \sim \mathcal{D}_{x.}; y_w \sim \pi_\theta(y|x_{\text{short}})} |\log \pi_\theta(y_w \mid x_{\text{short}}) - \log \pi_\theta(y_w \mid x_{\text{long}})|. \tag{10}$$

Here, we employ the absolute function to ensure non-negative training loss. From the reward expressions in Table 1, one can observe that SoLo-KL also promotes the convergence between $r_\phi(x_{\text{short}}, y_w)$ and $r_\phi(x_{\text{long}}, y_w)$. For DPO and SimPO, SoLo-KL is equivalent to SoLo-RA, as the reward coefficients $\beta$ can be integrated into the coefficient $\alpha$ in Eq. (9). As shown in Figure 2b, for DPO and SimPO, the performance of SoLo-KL and SoLo-RA is comparable, with minor differences attributed to the slight variations in $\alpha$. However, for ORPO, SoLo-RA significantly outperforms SoLo-KL on both benchmarks, demonstrating the effectiveness of direct reward alignment.

**Decoupling long-context alignment yields better results.** As shown in Figure 2b, SoLoPO outperforms Expand-Long-PO, a non-decoupled approach across different PO algorithms. We posit that the decoupled approach, through SoLo-RA, explicitly improves the model's contextual knowledge utilization ability by enabling direct comparison between short and long contexts, while explicitly strengthening the model's perception of rewards and preferences. We further test the contextual knowledge utilization ability of different models on *NIAH-Plus* [84] to validate the above claim. As shown in Table 5, SoLoPO achieves consistently greater improvements over Expand-Long-ORPO, confirming the rationality and effectiveness of our decoupled approach. Similar trends can be observed in the results of DPO and SimPO presented in Table

Table 5: Performance gains of various ORPO over Instruct model on *NIAH-Plus* [84].

| Model | S-Doc QA | M-Doc QA | AVG. |
|---|---|---|---|
| **Qwen2.5-7B-Instruct** | | | |
| $M_{\text{expand-long}}^{\text{ORPO}}$ | 23.94 | 17.29 | 20.62 |
| $M_{\text{SoLo}}^{\text{ORPO}}$ | **25.98** | **18.83** | **22.41** |
| **LLama3.1-8B-Instruct** | | | |
| $M_{\text{expand-long}}^{\text{ORPO}}$ | 11.65 | 9.77 | 10.71 |
| $M_{\text{SoLo}}^{\text{ORPO}}$ | **11.82** | **20.16** | **15.99** |

9. Additionally, Figure 9 in Appendix G presents heatmaps of model performance on *NIAH-Plus* when trained with different ORPO variants. Compared to Long-ORPO or Expand-Long ORPO, SoLo-ORPO significantly enhances the model's ability to retrieve information across various depths and context lengths in both single-hop and multi-hop settings, which further supports our claim.

**Impact of reward alignment coefficient $\alpha$ in SoLoPO** To evaluate the influence of $\alpha$ in Eq. (9), we progressively adjust $\alpha$ in SoLo-ORPO across two distinct foundation models. As shown in Figure 3, all architectures exhibit characteristic response curves with clear peaks in performance metrics, exceeding Long-ORPO in most settings. See Appendix H.4 for analysis of DPO and SimPO.

### 4.3 EFFICIENCY ADVANTAGE OF SOLOPO

The chosen-only SoLo-RA reduces the processing of long texts during training, thereby improving overall efficiency. As illustrated in Figure 4, we fix the length of $x_{\text{short}}$ at $1K$ and investigate how varying lengths of $x_{\text{long}}$ affect the performance gains and the efficiency gains in training time for SoLo-ORPO in the Qwen2.5-7B setting using $4\times$A100 GPUs. Results show that as the length of $x_{\text{long}}$ increases, SoLo-ORPO achieves significant efficiency gains over the vanilla ORPO, cutting run time by 42% and 39% for $x_{\text{long}}$'s lengths of $8K$ and $16K$, respectively. Moreover, for Qwen2.5-7B with a $32K$ pretrained context size, setting the length of $x_{\text{short}}$ and $x_{\text{long}}$ to $1K$ and $8K$, respectively, yields a favorable trade-off between model performance and computational efficiency. Notably, as shown in Figure 1b, with only ZeRO stage 3 and offloading enabled, SoLoPO supports trainable

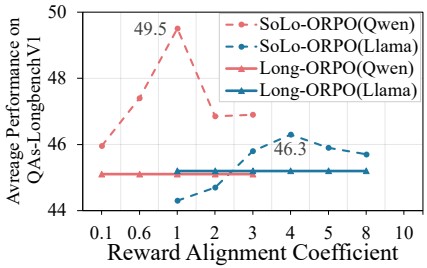 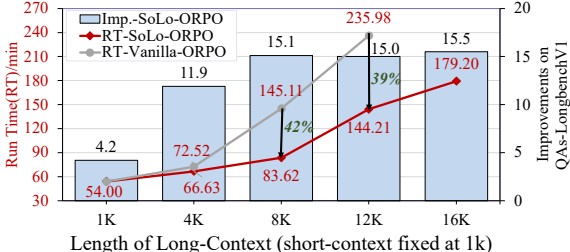

Figure 3: Performance w/ different $\alpha$ in SoLo-ORPO.

Figure 4: Run time (RT) and performance gains (Imp.) under varying lengths of $x_{\text{long}}$, with $x_{\text{short}}$ fixed at $1K$.

length up to $19K$ tokens, while vanilla methods are limited to $9K$. See Appendix H.5 for more experimental details and discussions.

## 5 RELATED WORK

**Long-Context Data Augmentation**   This approach leverages advanced LLMs to synthesize high-quality, long-dependency instruction-following data, used with PO or SFT for long-context alignment [43, 80, 91, 75, 4]. However, as text length increases, it becomes less reliable and efficient [63]. Moreover, short-context alignment methods may underperform in long-context settings [15, 18, 37].

**Long-Context Alignment Objective Optimization**   Fang et al. [18] propose LongPPL and LongCE loss to identify key tokens in long-text modeling and increase the loss weights for these critical tokens, thereby improving the effectiveness of long-context SFT. LongPO [11] searches for preference pairs based on short texts and applies them to long-context DPO training to achieve the non-decoupled short-to-long alignment discussed in our paper. Additionally, LongPO introduces a short-to-long constraint, replacing $\pi_{ref}(y \mid x_{\text{long}})$ with $\pi_{ref}(y \mid x_{\text{short}})$, thereby maintaining the short-context ability. However, LongPO focuses on context size expansion while its optimization is restricted to DPO. Both LongPO and LongCE suffer from limited training efficiency due to their reliance on long text processing, with LongCE incurring additional overhead associated with critical token detection.

SoLoPO introduces a theory-based decoupling strategy for long-context preference optimization, enabling more effective modeling of contextual knowledge localization and reasoning. The *chosen-only* SoLo-RA variant improves performance while facilitating data construction and training efficiency. Moreover, integrating SoLoPO with long-context data augmentation may further improve its alignment performance. A more comprehensive review of related work is provided in Appendix B.

## 6 CONCLUSION

In this work, we propose SoLoPO, a general framework for long-context preference optimization (PO). Our method decouples long-context PO into short-context PO and short-to-long reward alignment, supported by both theoretical and empirical evidence. Experimental results demonstrate that the *chosen-only* variant of SoLoPO consistently outperforms vanilla PO methods and enhances the model's generalization ability in handling long contexts across diverse domains and lengths, while significantly improving both data and training efficiency. SoLoPO highlights the importance of the connection between short and long texts, paving the way for more effective long-context alignment.

Our findings open up several promising avenues for future investigation, such as enhancing training efficiency for fully context-relevant tasks and exploring how the core ideas of SoLoPO can inform the optimization of long-output generation tasks. A more detailed discussion is provided in Appendix C.

## ACKNOWLEDGMENTS

Yang Gao was supported by the Major Research Plan of the National Natural Science Foundation of China (Grant No. 92370110) and the Joint Funds of the National Natural Science Foundation of China (Grant No. U21B2009).

ETHICS STATEMENT

This work focuses on optimizing preference learning objectives for large language models (LLMs) in long-context scenarios. It does not introduce unexpected ethical risks beyond those commonly considered in standard NLP research. Although LLMs are trained on large amounts of Internet text that may contain harmful content, our study targets long-context understanding rather than direct deployment, which greatly reduces the risk of propagating biased information. All models and datasets used in our experiments are open-sourced and publicly available, ensuring transparency and minimizing potential ethical concerns.

REPRODUCIBILITY STATEMENT

We have provided comprehensive details of our method (Section 2.2 and Appendix I), data synthesis pipeline (Section D), model training settings (Appendix E.1), and evaluation benchmarks and configurations (Appendix E.2) in the main paper. All datasets and models used in this work are openly available, with direct access links provided in our paper and the supplementary materials. To further facilitate reproduction of our results, we release the complete source code, data examples, and step-by-step usage instructions in the supplementary materials. These resources are intended to enable other researchers to fully replicate and verify our experiments.

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

## A  THE USE OF LARGE LANGUAGE MODELS

We explicitly disclose that large language models (LLMs) were used solely for the following purposes: (1) **Writing refinement** – limited to minor grammar correction, wording improvement, and stylistic polishing of the manuscript text; (2) **Data generation** – specifically, the preference data required for our experiments were sampled from the corresponding open-source Instruct models, following the procedures described in the paper.

LLMs were not used for idea conception, methodological design, result analysis, or any other substantive scientific contribution to this work. All research ideas, methodological innovations, experimental executions, analyses, and conclusions were conceived, implemented, and validated entirely by the authors.

## B  RELATED WORK

Numerous studies focus on extending the limited pre-training context windows of LLMs to support longer inputs, including post-training on long-context corpora [20, 12, 81], designing novel architectures [24, 46, 23], or modifying positional encodings [51, 26, 70, 88]. However, researches reveal that the capability within the pretraining window of current LLMs has not been fully activated, resulting in suboptimal performance on long-context tasks [31, 36, 8, 79]. To address this challenge, existing approaches primarily focus on two aspects: data augmentation and training objective optimization.

**Long-Context Alignment based on Data Augmentation**  Most research [43, 80, 91, 75] synthesizes high-quality, long-dependency instruction-following data for supervised fine-tuning or offline preference optimization. Instruction synthesis [43, 4] directly leverages one or multiple real long documents to prompt advanced LLMs to generate diverse instructions and responses for long-text alignment. Context synthesis [91], on the contrary, is built on real QA data by having models synthesize background contexts based on questions, then randomly concatenates multiple synthetic contexts to create long-context instruction-following data. Although data augmentation demonstrates some effectiveness, directly applying short-context training methods to the long-context setting may overlook differences between short and long contexts, thus failing to fully activate the model's potential capabilities [15, 37]. While SoLoPO incorporates the connection between short and long contexts into its training objective, it can be combined with data enhancement techniques, which has the potential to further enhance model performance.

**Long-Context Alignment based on Training Objective Optimization**  Some works explore optimizing training objectives to further enhance long-context capabilities in LLMs. Fang et al. [18] propose LongCE loss to identify key tokens in long-text modeling and increase the loss weights for these critical tokens, thereby improving the effectiveness of long-context SFT. LOGO [63] employs multiple negative samples and adapts the SimPO objective to minimize the probability of generating various dis-preference instances. LongPO [11] searches for preference pairs based on short texts and applies them to long-context DPO training to achieve the non-decoupled short-to-long alignment discussed in our paper. Additionally, LongPO introduces a short-to-long constraint, utilizing the output distribution of short texts on the reference model as a reference during long-context DPO training (replacing $\pi_{ref}(y \mid x_{long})$ with $\pi_{ref}(y \mid x_{short})$), thereby maintaining short-context capabilities. Although LOGO and LongPO adopt similar data construction strategies to ours, they fundamentally differ in that they do not decouple the optimization objectives. As a result, these methods fall into the category of non-decoupled short-to-long alignment discussed in our work. Moreover, LOGO, LongPO and LongCE suffer from limited training efficiency due to their reliance on long text processing, with LongCE incurring extra overhead from critical token detection.

SoLoPO introduces a theoretically grounded framework for long-context preference optimization. Specifically, SoLoPO explicitly models the connection between short and long contexts, decoupling long-text preference optimization into short-text preference optimization and short-to-long reward alignment. The *chosen-only* variant of SoLoPO not only improves the model's long-context ability, but also significantly enhances the efficiency of both data construction and training procedure.

## C  LIMITATIONS & FUTURE WORK

**Training Efficiency Enhancements**   For tasks where compression rate is 100%, such as long-context machine translation, SoLoPO is equivalent to the original PO algorithm, offering no gain in training efficiency. Given that redundancy also exists in the hidden states of LLMs [32, 50, 41], future research could extend token-level compression to hidden-state-level compression, potentially by combining our approach with KV-cache compression techniques [42, 41]. Such an extension would better support a wider variety of long-text applications. Moreover, although SoLoPO leverages the chosen-only SoLoRA strategy, it remains necessary to process long sequences, which can lead to efficiency bottlenecks when dealing with large-scale datasets. Future work could explore the decoupling strategy of SoLoPO in combination with data pruning techniques [53, 47], aiming to appropriately reduce the processing of long-context inputs and thereby improve training efficiency.

**Toward Extended Theoretical Analysis**   SoLoPO is primarily designed for long-context *input* scenarios, and therefore does not directly address the challenges of long-text *generation* [7]. Extending our theoretical analysis to long-text generation settings represents a natural and important direction for future work, which would further broaden the applicability of SoLoPO. The core principle of SoLoPO posits conditional equivalence between short and long inputs. We believe that this concept similarly extends to long-output generation tasks, as exemplified below:

- Long Chain-of-Thought (CoT) [61]: A long CoT that ultimately yields a correct final answer is equivalent to a concise CoT achieving the same result from a task completion perspective.
- Text Refinement: A stylistically refined and thus longer text can be deemed semantically equivalent to a plainer, shorter version, as long as the core semantic meaning is retained.
- Story Generation [25]: Longer story outputs are functionally equivalent to shorter versions if both fulfill a user-specified narrative arc or core plot, despite offering greater descriptive depth.

Moreover, as observed from Equation (9), original preference optimization (short-context PO) focuses on discrepancies in the output space, whereas our proposed SoLo-RA emphasizes relationships in the input space. This raises an intriguing question: *Can the decoupled preference modeling approach underlying SoLoPO be generalized to other tasks where modeling input-side connections plays a critical role?* (other context-aware tasks, such as complex instruction following [83] and context-faithful alignment [9].) Investigating this direction may yield new insights into the design of more expressive and flexible preference optimization frameworks.

**More experimental analysis**   Due to resource limitations, our current experiments primarily focus on efficiently activating capabilities within the model's pretraining context window ($32K$). Future work should further evaluate the effectiveness of SoLoPO on even longer context and larger model scales to fully understand its capabilities. Additionally, SoLoPO introduces two hyperparameters—compression ratio $c$ and reward alignment coefficient $\alpha$—which require manual tuning. Future work could explore automated methods for determining optimal values for these parameters. Moreover, while we believe that SoLoPO could support self-evolving mechanisms for progressive context window expansion [11, 63], this remains to be validated via more comprehensive analysis.

**Higher-Quality Data Synthesis**   While our current approach to constructing the short-to-long dataset is simple yet effective, it suffers from limited realism and diversity. Future work could explore integrating SoLoPO with existing data augmentation techniques to synthesize more realistic long-context instruction-following data, such as instruction or context synthesis grounded in authentic data sources [43, 91, 75, 4]. Moreover, extending the dataset to cover a broader range of long-text scenarios—such as long-document summarization, long-in-context learning, and long-form dialogue understanding—could provide a more comprehensive improvement of models' long-context processing capability.

# D  DATASET CONSTRUCTION

In this section, we present the methodology for the short-to-long preference dataset construction. As noted in our preliminary experiments and related studies [91, 22], training on shorter texts can still yield improvements in performance over longer contexts. Inspired by these findings, we heuristically set the average length of short contexts ($x_{short}$) to approximately $1K$ tokens and long contexts ($x_{long}$) to around $8K$ tokens, which corresponds to 25% of the pretraining context window ($32K$) of Qwen2.5-7B-Instruct. The overall data construction pipeline is illustrated in Figure 5.

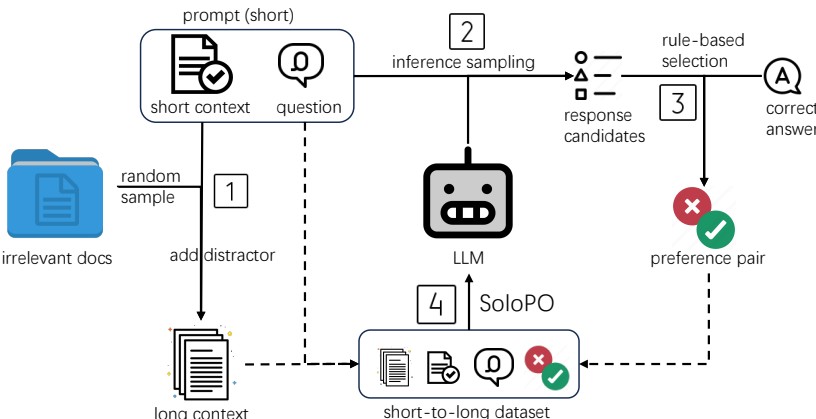

Figure 5: Illustration of the construction pipeline for the short-to-long dataset. (1) Irrelevant documents are randomly sampled and concatenated with the original short input to form long contexts. (2) Multiple candidate responses are generated based on the short context and question via the instruct model. (3) Preference pairs are curated using a sub-em[2]based selection guided by ground-truth answers. (4) The final short-to-long dataset, composed of short contexts, long contexts, questions, and preference pairs, is used for training LLM with SoLoPO.

**Context Synthesis**  We follow the strategy proposed by RULER [31] to generate synthetic contexts based on the MuSiQue dataset. Specifically, given an original sample $(q, a, D_{supporting})$, where $q$ and $a$ denote the question and ground-truth answer respectively, and $D_{supporting}$ is the set of supporting evidence documents, we synthesize a context of target length L as follows: we randomly sample a set of irrelevant documents $D_{irrelevant}$, such that the total token count of $D_{supporting} \cup D_{irrelevant}$ approximates $L$. The token count is calculated using the tokenizer of the corresponding instruct model. In this setup, both short and long contexts contain the necessary information to answer the question; however, the long contexts include more distractor content, thereby simulating the redundancy in natural language.

**Preference Pairs Construction**  To evaluate the effectiveness of short-to-long alignment compared with original long-context alignment, we construct preference pairs based on both short and long contexts using instruct models. Specifically, for each input context and question, we first generate $N = 32$ Chain-of-Thought responses using a sampling temperature of $0.85$ to encourage response diversity. We then apply the sub-em method to identify the chosen and rejected responses within the generated samples. A final preference pair is formed by randomly selecting one from the chosen and the rejected candidates, respectively. Samples where all responses are correct/incorrect are discarded.

To enhance the validity of the comparative analysis, we prioritize constructing training examples from the intersection of samples that can be successfully constructed using both short and long contexts. This ensures that differences in model performance stem primarily from alignment strategies rather than data distribution. The final size of each train set is 5000, as detials shown in Table 2 and an example is shown in Table 6. The prompt template used for data construction is illustrated in Figure 6.

---

[2]Alternative evaluation methods, such as LLM-as-Judge, may be employed provided they can differentiate preference pairs.

Table 6: An example preference pair sampled from Qwen2.5-7B-Instruct using the short context ($x_{short}$). Certain reasoning details have been omitted and denoted by "…" for conciseness.

| | |
|---|---|
| **question** | `When was the institute that owned The Collegian founded?` |
| **answer** | `1960` |
| **chosen** | `To find when the institute that owned The Collegian was founded,`
`let's follow these steps:`
`1.  Identify the owner of The Collegian:...`
`2.  Find the founding year of Houston Baptist University:  ...`
`The answer is:  1960` |
| **rejected** | `To answer the question, we need to identify which institute`
`"The Collegian" is associated with and then find its founding date.`
`However, the provided passages do not explicitly link The Collegian ...`
`no founding date for the university is given in the passages provided.`
`The answer is:  No answer.` |

Write an accurate and concise answer for the given question using only the provided search results (some of which might be irrelevant). Start with an accurate, engaging, and concise explanation based only on the provided documents. Must end with 'The answer is: {{your final answer}}'. Use an unbiased and journalistic tone. If the question cannot be answered, end with 'The answer is: No answer'.
Input:
Passage "[TITLE-1]":
[DOCUMENT-1]
Passage "[TITLE-2]":
[DOCUMENT-2]
…
Question: [QUERY] Think step-by-step.

Answer:

Figure 6: Prompt template used for data construction and training, adapted from Li et al. [40]

## E    IMPLEMENTATION DETAILS

In this section, we describe the implementation details of our experiments, including the configurations for model training and evaluation.

### E.1    MODEL TRAINING CONFIGURATION

**General training settings**    We train our model using LLaMAFactory [86] for data with a maximum length $\leq 8K$. To enhance training efficiency and better utilize GPU memory, we employ Flash-Attention 2 [13] and DeepSpeed ZeRO stage 3 with offloading [55] strategies. All models are fully trained on four NVIDIA A100 80GB GPUs in bf16 precision. We use the AdamW optimizer [45] together with a cosine learning rate scheduler. The `warmup_ratio` is set to 0.1 and the `total batch size` is 64. For a fair comparison, for each method we select the checkpoint that achieves the best performance on the QA tasks in LongBench-V1 during a single training epoch[3] as the final model to be evaluated across different benchmarks.

**Supervised fine-tuning (SFT)**    For SFT, the maximum learning rate is set to $1 \times 10^{-5}$.

**Original preference optimization**    We apply DPO, SimPO, and ORPO with a smaller maximum learning rate $1 \times 10^{-6}$. Following the original methods [54, 48, 30], for DPO and ORPO, we set the $\beta = 0.1$, and for SimPO, we set $\beta = 2.0$ and $\gamma = 1.4$.

**Short-to-Long preference optimization (SoLo-PO)**    For SoLo-PO, we maintain the same training parameters as in the original method. Specifically, SoLo-DPO, SoLo-SimPO, and SoLo-ORPO achieve optimal performance on LongBenchV1 with the reward alignment coefficient $\alpha$ in Equation (9) set as 3, 1, and 1, respectively, for Qwen2.5-7B, while SoLo-ORPO yields better performance with $\alpha = 4$ for LLaMA3.1-8B.

---

[3]Results show that, on our dataset, all training methods achieve their best performance within a single epoch.

**LongPO**  We train Qwen2.5-7B-Instruct on our constructed short-to-long dataset with the publicly available LongPO codebase[4], using the default hyperparameter settings, including the optimizer, learning rate and the learning rate scheduler, except for the batch size, which is set to 64 to match our setup. In addition, we also evaluate the publicly released model checkpoint[5] for direct comparison.

### E.2 EVALUATION DETAILS

#### E.2.1 EVALUATION BENCHMARKS

To comprehensively evaluate the capabilities of SoLoPO, we conduct experiments across a diverse set of benchmark datasets, as follows:

**QA tasks from LongBenchV1 and RULER**  Given that our training data is derived from the multi-hop QA dataset MuSiQue [64], with a maximum sequence length of $8K$ tokens, we primarily assess SoLoPO's performance on long-context QA tasks within a $32K$ context size. Specifically, we use the QA tasks from LongBenchV1 [5] to evaluate the model's generalization across various real-world domains. These include single-document QA tasks such as Qasper [14], NarrativeQA [35], and MultiFieldQA-En [5], as well as multi-document QA tasks including HotpotQA [78], MuSiQue [64], and 2WikiMQA [29]. Additionally, we incorporate synthetic QA tasks from RULER—SquadQA [56], HotpotQA [78], and MuSiQue [64]—at varying context lengths ($4K, 8K, 16K$, and $32K$ tokens)—to further analyze the model's length extrapolation abilities.

**LongBenchV2**  To explore the potential of SoLoPO in more diverse and longer-context scenarios, we further evaluate it on the full suite of tasks in LongBenchV2 [6]. This benchmark covers a wide range of long-context tasks, including question answering, abstractive summarization, and in-context learning, with input lengths spanning below $32K$ words, between $32K$ and $128K$ words, and beyond $128K$ words.

**Open LLM Leaderboard**  Prior works [72, 16, 11] note that aligning models for long-context tasks may forget their short-context capabilities. To assess short-context performance retention of different methods, we adopt evaluations from the Open LLM Leaderboard[6] [19]. These include widely used tasks such as MMLU-Pro [66], MATH [27], GPQA [57], IFEval [87], and BBH [60], which valuate general knowledge, mathematical reasoning, scientific (chemistry, biology, physics) knowledge, instruction following, and complex reasoning, respectively.

**NIAH-Plus**  As described in Section 4.2, to better understand how different training strategies affect the contextual knowledge utilization capability, we employ the NIAH-Plus [84] benchmark. This needle-in-a-haystack QA benchmark includes both single-document and multi-document settings, and is designed to directly probe a model's capacity for context-aware retrieval and multi-step reasoning.

#### E.2.2 EVALUATION SETTINGS

**Prompts for Evaluation**  For QA tasks in LongBenchV1 and RULER, we use the same prompt template as employed during data construction and model training, which is illustrated in Figure 6. For all other benchmarks mentioned in this paper, we adopt their publicly released prompts. Specifically, as shown in Figure 7, for LongBenchV2, we employ a single-stage chain-of-thought prompting strategy to generate answers directly, differing from the official two-stage evaluation protocol. For the Open LLM Leaderboard and NIAH-Plus benchmarks, we follow the default prompts used in the official implementation code (lm-evaluation-harness[7] and NIAHaystack-PLUS[8] repositories).

**Decoding hyperparameters**  We use greedy decoding for evaluation on LongBenchV1, RULER, and NIAH-Plus. For other benchmarks, we follow the official decoding settings with temperature

---

[4]`https://github.com/DAMO-NLP-SG/LongPO`
[5]`https://huggingface.co/DAMO-NLP-SG/Qwen2.5-7B-LongPO-128K`
[6]`https://huggingface.co/spaces/open-llm-leaderboard/open_llm_leaderboard`
[7]`https://github.com/EleutherAI/lm-evaluation-harness`
[8]`https://github.com/zuucan/NeedleInAHaystack-PLUS`

> Please read the following text and answer the question below.
>
> <text>
> [CONTEXT]
> </text>
>
> What is the correct answer to this question: [QUESTION]
> Choices:
> (A) [CHOICE_A]
> (B) [CHOICE_B]
> (C) [CHOICE_C]
> (D) [CHOICE_D]
>
> Let's think step by step. And format your final answer choice as follows: "The correct answer is (insert answer here)".

Figure 7: Prompt template used for LongBenchV2 evaluation, adapted from Bai et al. [6]

values of 0.1 and 0 for LongBenchV2 and the Open LLM Leaderboard, respectively. The error analysis for LongbenchV2 is provided in Appendix F.

## F    STANDARD DEVIATION OF LONGBENCHV2

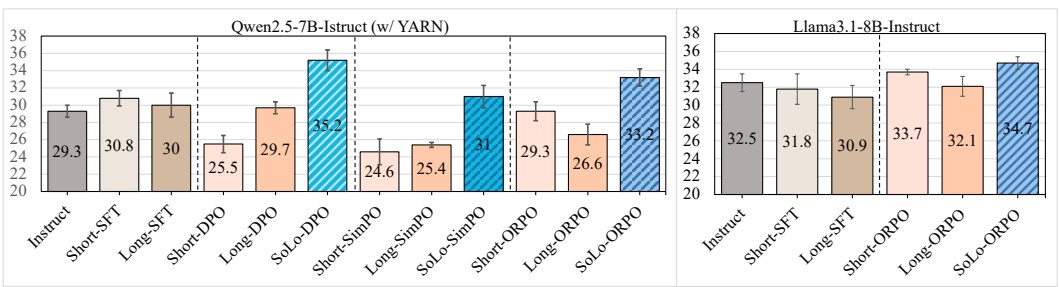

Figure 8: Overall Performance on LongbenchV2. **1.** We report the average score with standard deviation across 5 evaluation runs for each model. **2.** All of these metrics are reasonable.

As mentioned in Appendix E.2, we use greedy decoding (temperature $= 0$) for all benchmarks except LongbenchV2, where the temperature is set to $0.1$. Here, we report the standard deviation of LongbenchV2 in Figure 8 and Table 4, where all of these metrics are reasonable.

## G    MORE DETAILED BENCHMARK RESULTS

We present more detailed results for our experiments presented in Section 4.

**Detailed results on QA tasks from LongbenchV1 and RULER**    Since the training dataset we use, MuSiQue, is designed for multi-hop QA tasks, in Section 4.1, we primarily evaluate various methods on the QA tasks from LongBenchV1 and RULER. This allows us to examine model performance on real-world scenarios and across varying input lengths. Results are presented in Table 7 and Table 8.

**More detailed results on NIAH-Plus**    In Section 4.2, to validate whether the decoupled approach may more effectively enhance the model's ability to locate contextual knowledge, we evaluate Expand-Long-PO (a non-decoupled approach) and SoLoPO (our decoupled approach) on NIAH-Plus with full results presented in Table 9. SoLoPO consistently outperforms Expand-Long-PO across all evaluation scenarios and preference optimization algorithms, demonstrating the effectiveness of our decoupling-based short-to-long preference optimization approach. Figure 9 presents heatmaps of the performance gains of various ORPO over Qwen2.5-7B. SoLo-ORPO, powered by the SoLoPO's decoupling strategy, surpasses Long-ORPO and Expand-Long ORPO on NIAH-Plus. It achieves superior information retrieval and utilization across various depths and context lengths in both single-hop and multi-hop settings, fundamentally improving contextual knowledge localization.

Table 7: Detailed Results of QA tasks from LongBenchV1

| Model | Single-Doc QA | | | | Multi-Doc QA | | | | AVG. |
|---|---|---|---|---|---|---|---|---|---|
| | NarrativeQA | Qasper | MultiFieldQA-En | Avg. | HotpotQA | 2WikiMQA | MuSiQue | Avg. | |
| Qwen2.5-7B-Instruct | | | | | | | | | |
| Instruct | 15.8 | 31.3 | 41.0 | 29.4 | 39.6 | 48.9 | 29.7 | 39.4 | 34.4 |
| $M^{\text{SFT}}_{\text{short}}$ | 13.4 | 34.0 | 39.3 | 28.9 | 47.3 | 63.3 | 34.5 | 48.4 | 38.6 |
| $M^{\text{SFT}}_{\text{long}}$ | 21.3 | 39.4 | 43.7 | 34.8 | 55.4 | 65.7 | 46.5 | 55.8 | 45.3 |
| $M^{\text{DPO}}_{\text{short}}$ | 17.2 | 41.3 | 45.4 | 34.6 | 50.9 | 68.0 | 36.4 | 51.8 | 43.2 |
| $M^{\text{DPO}}_{\text{long}}$ | 22.6 | 41.1 | 43.4 | 35.7 | 60.2 | 66.3 | 48.0 | 58.2 | 46.9 |
| $M^{\text{SoLo}}_{\text{DPO}}$ | **26.1** | **43.0** | 44.8 | 38.0 | 58.8 | 63.3 | **50.7** | 57.6 | 47.8 |
| $M^{\text{SimPO}}_{\text{short}}$ | 20.3 | 41.4 | 42.5 | 34.7 | 55.5 | 67.1 | 38.1 | 53.6 | 44.1 |
| $M^{\text{SimPO}}_{\text{long}}$ | 16.8 | 41.0 | 44.8 | 34.2 | 56.3 | 64.9 | 43.6 | 54.9 | 44.6 |
| $M^{\text{SoLo}}_{\text{SimPO}}$ | 25.5 | 42.8 | 46.0 | **38.1** | 61.5 | 68.4 | 46.0 | 58.6 | 48.4 |
| $M^{\text{ORPO}}_{\text{short}}$ | 13.4 | 34.0 | 39.3 | 28.9 | 47.3 | 63.3 | 34.5 | 48.4 | 38.6 |
| $M^{\text{ORPO}}_{\text{long}}$ | 21.3 | 39.4 | 43.7 | 34.8 | 55.4 | 65.7 | 46.5 | 55.8 | 45.3 |
| $M^{\text{SoLo}}_{\text{ORPO}}$ | 24.9 | 41.4 | **46.6** | 37.6 | **64.3** | **71.7** | 48.1 | **61.4** | **49.5** |
| LLama3.1-8B-Instruct | | | | | | | | | |
| Instruct | 12.3 | 37.3 | 41.5 | 30.4 | 54.8 | 59.8 | 33.4 | 49.3 | 39.8 |
| $M^{\text{SFT}}_{\text{short}}$ | 19.1 | 37.1 | 42.9 | 33.0 | **59.0** | 65.4 | 44.4 | 56.3 | 44.6 |
| $M^{\text{SFT}}_{\text{long}}$ | 20.3 | 39.6 | 45.1 | 35.0 | 56.6 | 68.3 | 47.1 | 57.3 | 46.1 |
| $M^{\text{ORPO}}_{\text{short}}$ | 17.1 | 38.8 | 43.3 | 33.1 | 55.4 | 67.1 | 42.7 | 55.1 | 44.1 |
| $M^{\text{ORPO}}_{\text{long}}$ | 19.6 | **40.1** | **46.7** | **35.4** | 57.2 | 68.3 | 40.8 | 55.4 | 45.4 |
| $M^{\text{SoLo}}_{\text{ORPO}}$ | **21.5** | 40.0 | 44.2 | 35.2 | 54.8 | **69.0** | **48.2** | 57.3 | **46.3** |

Table 8: Detailed Results of QA tasks from RULER

| Model | SquadQA(OOD) | | | | | HotpotQA(OOD) | | | | | MuSiQue (InDomain) | | | | | AVG. |
|---|---|---|---|---|---|---|---|---|---|---|---|---|---|---|---|---|
| | 4k | 8k | 16k | 32k | avg. | 4k. | 8k | 16k | 32k | avg. | 4k | 8k | 16k | 32k | avg. | |
| Qwen2.5-7B-Instruct | | | | | | | | | | | | | | | | |
| Instruct | 61.3 | 49.5 | 37.0 | 35.2 | 45.8 | 59.6 | 61.4 | 51.3 | 47.7 | 55.0 | 40.9 | 39.4 | 24.6 | 20.8 | 31.4 | 44.0 |
| $M^{\text{SFT}}_{\text{short}}$ | 72.3 | 57.5 | 41.4 | 26.5 | 49.4 | 68.2 | 66.0 | 52.6 | 44.8 | 57.9 | 50.8 | 46.5 | 33.0 | 24.2 | 38.6 | 48.7 |
| $M^{\text{SFT}}_{\text{long}}$ | 70.9 | 64.4 | 61.8 | 54.1 | 62.8 | 71.0 | 67.7 | 68.4 | 66.7 | 68.5 | 55.9 | 52.1 | 44.9 | 36.5 | 47.3 | 59.5 |
| $M^{\text{DPO}}_{\text{short}}$ | **76.2** | 64.0 | 41.9 | 47.8 | 57.5 | 74.4 | 72.2 | 58.9 | 59.7 | 66.3 | 62.1 | 53.7 | 35.0 | 33.1 | 46.0 | 56.6 |
| $M^{\text{DPO}}_{\text{long}}$ | 75.6 | 66.7 | 62.5 | 52.7 | 64.4 | 74.6 | 69.8 | 70.9 | 67.0 | 70.6 | 62.7 | 56.3 | 48.5 | 39.8 | 51.8 | 62.2 |
| $M^{\text{SoLo}}_{\text{DPO}}$ | 70.7 | 64.4 | 64.8 | **62.0** | 65.5 | 70.5 | 73.2 | 71.5 | 67.3 | 70.6 | 58.2 | 55.8 | 51.7 | 44.0 | 52.4 | 62.8 |
| $M^{\text{SimPO}}_{\text{short}}$ | 75.8 | 62.7 | 39.4 | 50.0 | 57.0 | **74.8** | 70.1 | 53.8 | 62.6 | 65.3 | 61.7 | 54.9 | 33.2 | 33.8 | 45.9 | 56.0 |
| $M^{\text{SimPO}}_{\text{long}}$ | 74.0 | 64.0 | 45.0 | 46.4 | 57.4 | 74.1 | 73.8 | 62.7 | 61.1 | 67.9 | 61.2 | 54.7 | 39.6 | 34.0 | 47.4 | 57.5 |
| $M^{\text{SoLo}}_{\text{SimPO}}$ | 75.2 | 67.6 | **66.9** | 60.6 | 67.6 | 74.3 | 73.9 | **71.8** | 69.9 | **72.5** | 58.1 | 56.5 | 49.3 | 42.9 | 51.7 | 63.9 |
| $M^{\text{ORPO}}_{\text{short}}$ | 75.2 | 64.3 | 48.8 | 45.8 | 58.5 | 73.3 | 69.8 | 62.8 | 61.5 | 66.9 | 59.0 | 52.1 | 40.8 | 32.5 | 46.1 | 57.1 |
| $M^{\text{ORPO}}_{\text{long}}$ | 70.5 | 61.2 | 48.4 | 45.9 | 56.5 | 68.7 | 69.0 | 63.4 | 59.5 | 65.2 | 55.4 | 49.5 | 38.1 | 31.4 | 43.6 | 55.1 |
| $M^{\text{SoLo}}_{\text{ORPO}}$ | 74.8 | **67.8** | 65.6 | 59.2 | 66.9 | 73.3 | **74.8** | 70.7 | 67.2 | 71.5 | **64.2** | **62.4** | **55.7** | **45.6** | **57.0** | **65.1** |
| LLama3.1-8B-Instruct | | | | | | | | | | | | | | | | |
| Instruct | 67.4 | 53.8 | 44.9 | 41.7 | 52.0 | 68.1 | 61.4 | 57.3 | 49.9 | 59.2 | 39.4 | 32.5 | 26.3 | 15.1 | 28.3 | 46.5 |
| $M^{\text{SFT}}_{\text{short}}$ | **70.7** | **66.0** | 61.7 | 55.4 | 63.5 | 67.6 | **67.8** | 66.2 | 62.5 | 66.0 | **56.7** | 49.1 | 47.5 | 38.6 | 48.0 | 59.2 |
| $M^{\text{SFT}}_{\text{long}}$ | 69.4 | 63.6 | 64.6 | 58.9 | 64.1 | 66.6 | 64.6 | 63.7 | 62.0 | 64.2 | 55.0 | 48.9 | 44.3 | 39.5 | 46.9 | 58.4 |
| $M^{\text{ORPO}}_{\text{short}}$ | 70.0 | 65.5 | **67.7** | 52.0 | 63.8 | 67.9 | 64.8 | 65.6 | 63.0 | 65.3 | 54.2 | 47.7 | 45.7 | 36.2 | 46.0 | 58.4 |
| $M^{\text{ORPO}}_{\text{long}}$ | 68.5 | 65.1 | 64.9 | 59.2 | 64.4 | 68.5 | 66.7 | **66.7** | 61.4 | 65.8 | 52.7 | 48.5 | 45.1 | 38.9 | 46.3 | 58.9 |
| $M^{\text{SoLo}}_{\text{ORPO}}$ | 67.0 | 63.5 | 64.5 | 59.8 | 63.7 | **70.3** | 67.0 | 66.0 | **66.8** | **67.5** | 55.8 | **57.1** | **49.8** | **48.1** | **52.7** | **61.3** |

## H MORE EXPERIMENTAL ANALYSIS

### H.1 MORE ANALYSIS ABOUT SOLO-DPO/SIMPO ON LONGBENCHV2

Table 4 presents results of Qwen2.5-7B (w/ YARN [51]) on LongBenchV2. For Qwen2.5-7B (w/ YARN), SoLoPO consistently outperforms original PO methods across all evaluation context lengths and difficulty levels. Specifically, (1) SoLo-ORPO also surpasses vanilla PO across all dimensions; (2) SoLo-DPO achieves the best overall performance, particularly on contexts $\geq 32$ and hard samples, likely due to the reference model ensuring better generalization; (3) SoLo-SimPO shows relatively weaker performance, possibly due to its reward model relies on normalized prediction probabilities, which can underperform on long-context evaluations like perplexity observed by Fang et al. [18].

Table 9: Comparative performance of SoLoPO and Expand-Long-PO on *NIAH-Plus* (within $128K$ context size). As shown in (↑), SoLoPO consistently outperforms Expand-Long-PO, validating our decoupled short-to-long preference optimization approach.

| Model | Single-document QA | Multi-document QA | AVG. |
|---|---|---|---|
| **Qwen-2.5-7B-Instruct** | | | |
| Instruct | 35.66 | 52.63 | 44.14 |
| $\overline{M}^{\text{DPO}}_{\text{expand-long}}$ | 55.98 | 68.02 | 62.00 |
| $M^{\text{DPO}}_{\text{SoLo}}$ | 59.35 (↑ 3.37) | 71.76 (↑ 3.74) | 65.56 (↑ 3.56) |
| $\overline{M}^{\text{SimPO}}_{\text{expand-long}}$ | 51.81 | 53.61 | 52.71 |
| $M^{\text{SimPO}}_{\text{SoLo}}$ | 60.85 (↑ 9.04) | **72.05** (↑ 18.44) | 66.45 (↑ 13.74) |
| $\overline{M}^{\text{ORPO}}_{\text{expand-long}}$ | 59.60 | 69.92 | 64.76 |
| $M^{\text{ORPO}}_{\text{SoLo}}$ | **61.64** (↑ 2.04) | 71.46 (↑ 1.54) | **66.55** (↑ 1.79) |
| **LLama3.1-8B-Instruct** | | | |
| Instruct | 32.04 | 32.8 | 32.42 |
| $\overline{M}^{\text{ORPO}}_{\text{expand-long}}$ | 43.69 | 42.57 | 43.13 |
| $M^{\text{ORPO}}_{\text{SoLo}}$ | **43.86** (↑ 0.17) | **52.96** (↑ 10.39) | **48.41** (↑ 5.28) |

(a) Single-document QA setting.

(b) Multi-document QA setting.

Figure 9: Comparison of performance improvements achieved by various ORPO methods relative to Qwen2.5-7B on the *NIAH-Plus* [84]. Expand-Long-ORPO and SoLoPO demonstrate significantly greater improvements than Long-ORPO, highlighting the effectiveness of short-to-long alignment. Moreover, SoLoPO provides greater gains than Expand-Long-ORPO, validating the effectiveness of our decoupled approach.

## H.2 TRAINING DYNAMICS OF DIFFERENT SoLo-RA APPROACHES (*chosen-only* VS. *both*)

As demonstrated in Figure 10a, the margin curves of the *both* SoLo-RA consistently lie beneath those of the *chosen-only* approach, with its ultimately converged margin values being substantially lower. This observation indicates that the *chosen-only* SoLo-RA exhibits superior fitting capability along the $x_{long}$ dimension. 10b reveals that during the initial optimization phase, the *both* SoLo-RA induces an anomalous transient increase in the probability of $p_\theta(y_l \mid x_{long})$, which should theoretically follow a monotonic decreasing trend. This suboptimal convergence behavior ultimately leads to significantly inferior optimization outcomes compared to the *chosen-only* approach as shown in Figure 2a.

## H.3 ABLATION EXPERIMENTS ON SoLo-RA

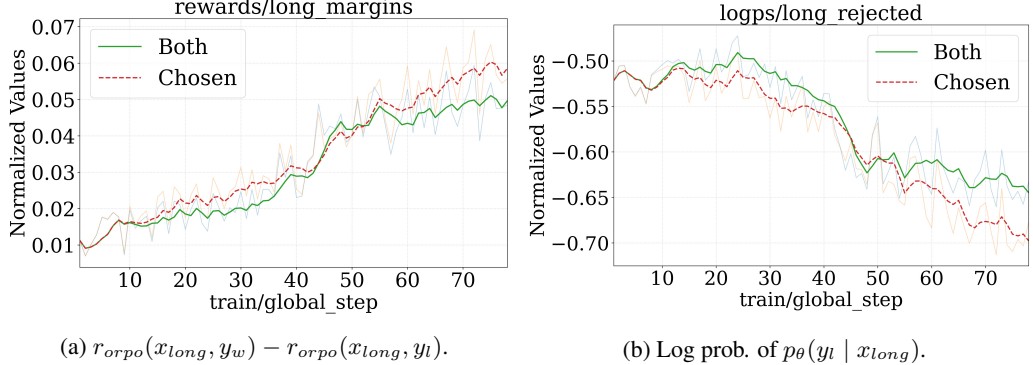

(a) $r_{orpo}(x_{long}, y_w) - r_{orpo}(x_{long}, y_l)$.

(b) Log prob. of $p_\theta(y_l \mid x_{long})$.

Figure 10: Changes of reward margins and log prob. of rejected response during SoLo-ORPO training given long contexts. Compared to the *both* approach, *chosen-only* short-to-long reward alignment achieves (a) larger reward margins, ensuring higher reward accuracy, while (b) simultaneously applying sufficient penalties to rejected responses, reducing their prediction probability. The chosen-only approach exhibits more precise reward modeling, ultimately achieving better alignment outcomes.

In Section 2.2, we hypothesize that SoLo-RA enhances the model's contextual knowledge localization ability (see Appendix I.9 for a detailed discussion). In this section, we conduct ablation studies on NIAH-Plus to further validate this hypothesis.

To conduct the ablation analysis of SoLo-RA, we focus on the performance of models trained with Short-PO, Expand-PO, and SoLoPO on NIAH-Plus, for the following reasons:

- Removing SoLo-RA from SoLoPO's optimization objective yields short-PO; however, this also eliminates the $x_{long}$ from the training data, making a direct comparison unable to disentangle the gains from data length on generalization [90].

- Expand-PO and SoLoPO are trained on identical data except for the absence of $x_{short}$ in Expand-PO, allowing their comparison to minimize potential confounding effects from training data length.

Table 10: Ablation studies for SoLo-RA on *NIAH-Plus* (within $128K$ context size).

| Model | S-Doc QA | M-Doc QA | AVG. |
|---|---|---|---|
| **Qwen-2.5-7B-Instruct** | | | |
| Instruct | 35.66 | 52.63 | 44.14 |
| $M_{short}^{DPO}$ | 51.25 | 64.70 | 57.98 |
| $M_{expand-long}^{DPO}$ | 55.98 | 68.02 | 62.00 |
| $M_{SoLo}^{DPO}$ | **59.35** | **71.76** | **65.56** |
| $M_{short}^{SimPO}$ | 50.84 | 63.59 | 57.22 |
| $M_{expand-long}^{SimPO}$ | 51.81 | 53.61 | 52.71 |
| $M_{SoLo}^{SimPO}$ | **60.85** | **72.05** | **66.45** |
| $M_{short}^{ORPO}$ | 45.02 | 59.17 | 52.10 |
| $M_{expand-long}^{ORPO}$ | 59.60 | 69.92 | 64.76 |
| $M_{SoLo}^{ORPO}$ | **61.64** | **71.46** | **66.55** |
| **LLama3.1-8B-Instruct** | | | |
| Instruct | 32.04 | 32.80 | 32.42 |
| $M_{short}^{ORPO}$ | 37.20 | 37.77 | 37.49 |
| $M_{expand-long}^{ORPO}$ | 43.69 | 42.57 | 43.13 |
| $M_{SoLo}^{ORPO}$ | **43.86** | **52.96** | **48.41** |

As shown in Table 10, SoLo-PO consistently outperforms both Short-PO and Expand-PO in contextual knowledge localization tasks with a 128K context length across different PO algorithms and base models. These results provide further evidence that SoLo-RA more effectively strengthens the capacity of the model to localize contextual knowledge.

### H.4 Impact of Reward Alignment Coefficient $\alpha$ in SoLo-DPO and SoLo-SimPO

Figure 11 shows the performance of SoLo-DPO and SoLo-SimPO under varying $\alpha$ in the Qwen2.5-7B setting. The results indicate that the optimal $\alpha$ values are 3 for SoLo-DPO and 1 for SoLo-SimPO, and in most cases SoLoPO outperforms Long-PO, suggesting its stability across different $\alpha$ values.

### H.5 Efficiency Analysis of SoLoPO

Thanks to the reduced overhead in handling long texts, SoLoPO offers notable advantages over the original algorithms in both computational efficiency and memory usage. In this section, we

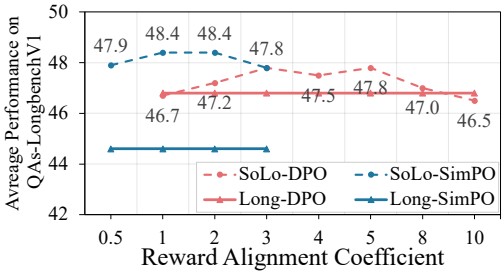

Figure 11: Performance with different $\alpha$ in SoLo-DPO and SoLo-SimPO in the Qwen2.5-7B setting. The optimal values of $\alpha$ for SoLo-DPO and SoLo-SimPO are 3 and 1, respectively.

present detailed empirical comparisons of runtime efficiency and provide a theoretical analysis of the computational speedup.

**Training Implementation Details** The experimental setup for Figure 1b adheres to the same training configuration described in Section E.1. For experiments in Section 4.3, we employ various optimization strategies of ZeRO [55] to maximize GPU memory utilization while enabling training on longer sequences. The specific training configurations are detailed in Table 11.

Table 11: Implementation details and run time of SoLo-ORPO and vanilla ORPO under varying lengths of $x_{long}$. **1.** the default configuration is Zero stage 3 without offloading **2.** "2-Stage Forward" indicates sequential forward passes for $(y_w, x_{long})$ and $(y_l, x_{long})$, as opposed to the default concatenated forward strategy. **3.** "OOM" denotes CUDA Out-of-Memory errors. **4.** SoLoPO significantly improves the training efficiency of the vanilla ORPO.

| Length | Method | Offloading | 2-Stage Forward for $x_{long}$ | Runtime (min) ↓ |
|---|---|---|---|---|
| $1K$ | vanilla | | | 54.00 |
| $4K$ | vanilla | | | 72.52 |
| | SoLo | | | 66.63 ($\downarrow 8\%$) |
| $8K$ | vanilla | ✓ | | 145.11 |
| | SoLo | ✓ | | 109.42 ($\downarrow 25\%$) |
| | SoLo | | | 83.62 ($\downarrow 42\%$) |
| $12K$ | vanilla | ✓ | ✓ | 235.98 |
| | SoLo | ✓ | | 144.21 ($\downarrow 39\%$) |
| $16K$ | vanilla | ✓ | ✓ | OOM |
| | SoLo | ✓ | | 179.20 |
| $19K$ | vanilla | ✓ | ✓ | OOM |
| | SoLo | ✓ | | 205.98 |

- **SoLoPO** When the lengths of $x_{long}$ ranging from $4K$ to $16K$ tokens, we employ a two-stage forward mechanism within LLaMAFactory [86] to perform SoLoPO training. Specifically, for the short-context PO, we apply the `concatenated_forward` [9] function directly on $(y_w, x_{short})$ and $(y_l, x_{short})$ to obtain $\log \pi_\theta(y_w \mid x_{short})$ and $\log \pi_\theta(y_l \mid x_{short})$. Subsequently, we conduct a separate forward pass over $(y_w, x_{long})$ to compute $\log \pi_\theta(y_l \mid x_{long})$. Finally, the SoLoPO loss is calculated based on the corresponding loss function in Table 1.

- **Vanilla PO** When the lengths of $x_{long}$ is less than or equal to $8K$ tokens, $(y_w, x_{long})$ and $(y_l, x_{long})$ can be efficiently processed using the `concatenated_forward` function, allowing for straightforward loss computation. However, at $x_{long}$ lengths of $12K$ tokens, the use of `concatenated_forward` leads to out-of-memory (OOM) errors. Thus, we adopt a 2-stage forward approach—processing $(y_w, x_{long})$ and $(y_l, x_{long})$ sequentially. For sequences as long as $16K$ tokens, even this serialized method becomes infeasible, necessitating the use of sequence parallelism techniques to enable training. For even longer $(x_{long})$ exceeding $16K$ tokens, only a

---

[9] https://github.com/hiyouga/LLaMA-Factory/blob/main/src/llamafactory/train/dpo/trainer.py#L179

sequence parallelism [33] training strategy becomes feasible. While these alternative approaches mitigate memory constraints, they inevitably increase overall training time.

**Computation speedup analysis of SoLoPO** Following Bai et al. [2], we define the compression rate $\rho \in (0, 1]$ as the ratio of the length of $x_{short}$ to that of $x_{long}$. Specifically, if the length of $x_{long}$ is denoted by $N$, then the length of $x_{short}$ is given by $\rho N$. We focus on the computation incurred by the policy model during training and ignore reference models. Since Transformer operations scale quadratically with sequence length (FLOPs $\propto n^2$), the total training computation for vanilla PO and SoLoPO can be expressed as:

$$\text{FLOPs}_{PO} = 2N^2, \quad \text{FLOPs}_{SoLoPO} = (2\rho^2 + 1)N^2. \tag{11}$$

Consequently, the computation speedup ratio of SoLoPO over vanilla PO can be expressed as:

$$\text{Speedup}(c) = \frac{\text{FLOPs}_{PO}}{\text{FLOPs}_{SoLoPO}} = \frac{2}{2\rho^2 + 1}, \quad \rho \in (0, 1]. \tag{12}$$

This indicates that SoLoPO achieves computational efficiency gains when $\rho < \frac{1}{\sqrt{2}} \approx 0.707$, *i.e.*, when $x_{short}$ is less than approximately 70.7% of the length of $x_{long}$.

**Potential Approaches for Optimizing Training Efficiency on High Compression Rate Tasks**
For tasks with high compression ratios, such as long-context machine translation ($\rho$=100%), the linguistic redundancy hypothesis underlying SoLoPO no longer holds. In this scenario, SoLoPO degrades to vanilla PO and offers no training efficiency gains. It is noteworthy that standard attention mechanisms in LLMs exhibit redundancy, which enables the application of KV compression [42] or sparse attention [71] techniques. We hypothesize that this model-inherent redundancy can be leveraged for representational compression (e.g., gist-token [82]). Consequently, for tasks lacking linguistic redundancy, the short-to-song paradigm could potentially be applied based on the model-inherent redundancy, thereby reducing training time and memory consumption for long-context scenarios.

Table 12: Performance of Qwen2.5-Instruct-14B trained with different methods on LongbenchV1-QAs and RULER-QAs. Across benchmarks, SoLoPO consistently outperforms vanilla PO algorithms.

| Model | QAs-LongBenchV1 | | | QAs-RULER | | | | | Run Time |
|---|---|---|---|---|---|---|---|---|---|
| | S-Doc QA | M-Doc QA | Avg. | 4k | 8k | 16k | 32k | Avg. | /min($\downarrow$) |
| 72B-Instruct | 37.80 | 61.10 | 49.40 | 65.70 | 64.4 | 61.20 | 55.0 | 61.60 | - |
| 14B-Instruct | 34.40 | 52.07 | 43.30 | 56.60 | 52.27 | 49.69 | 43.86 | 50.60 | - |
| $M_{short}^{SFT}$ | 36.20 | 59.13 | 47.70 | 64.47 | 59.46 | 55.67 | 46.31 | 56.48 | - |
| $M_{long}^{SFT}$ | 34.53 | 61.07 | 47.80 | 65.72 | 61.44 | 52.56 | 43.83 | 55.89 | - |
| $M_{short}^{DPO}$ | **37.83** | 65.33 | 51.60 | 71.32 | 65.79 | 63.12 | 55.97 | 64.05 | - |
| $M_{long}^{DPO}$ | 36.63 | 66.63 | 51.60 | **73.37** | 69.49 | 67.69 | 59.83 | 67.60 | 249 |
| $M_{SoLo}^{DPO}$ | 37.80 | **67.20** | **53.40** | 72.95 | **70.53** | **68.52** | **62.03** | **68.51** | **172** |
| $M_{short}^{SimPO}$ | 37.47 | 64.30 | 50.90 | 71.43 | 65.87 | 63.21 | 55.69 | 64.05 | - |
| $M_{long}^{SimPO}$ | 36.77 | **66.10** | 51.40 | 70.33 | 66.24 | 63.27 | 56.91 | 64.19 | 248 |
| $M_{SoLo}^{SimPO}$ | **38.40** | 65.67 | **52.00** | 71.79 | 67.85 | 66.52 | 60.47 | 66.66 | 169 |
| $M_{short}^{ORPO}$ | 39.37 | 63.80 | 51.60 | 68.63 | 63.85 | 61.47 | 52.96 | 61.73 | - |
| $M_{long}^{ORPO}$ | 38.80 | 62.73 | 50.80 | 68.69 | 64.19 | 62.16 | 52.83 | 61.97 | 248 |
| $M_{SoLo}^{ORPO}$ | **39.80** | **65.70** | **52.80** | 72.45 | 67.85 | 65.10 | 60.78 | 66.54 | 170 |

## H.6 ON THE SCALABILITY OF SOLOPO TO LARGER MODELS

To examine the scalability of SoLoPO to larger models, we conduct experiments on Qwen2.5-Instruct-14B with different preference optimization methods, constrained by available computational resources. The evaluation is performed on the primary benchmarks used in this paper, LongbenchV1-QAs and RULER-QAs. The training and evaluation settings follow those of Qwen2.5-Instruct-7B, except that the parameter $\alpha$ in SoLoPO is set to 1 and experiments are run on 8$\times$ H20 GPUs with `LLaMA-Factory(0.9.1)`. As shown in Table 12, SoLoPO consistently outperforms the original PO algorithms across different benchmarks while significantly improving training efficiency.

# I   THEORETICAL DERIVATION AND SUPPORTING ANALYSIS OF SOLOPO

In this section, we formulate the theoretical foundation for the short-to-long preference optimization (SoLoPO) method through a novel reward loss function, and develop a distance metric condition for applying the framework for generalized distance metrics. Furthermore, we systematically extend this framework to mainstream preference optimization paradigms, including Direct Preference Optimization (DPO), Simple Preference Optimization (SimPO) and so on, demonstrating its methodological generality.

## I.1   PROOF OF LEMMA 1

**Lemma 1.** *If the function $f(\cdot)$ of equation (4) is convex, then the following inequality holds true:*

$$
l_{\eta,\gamma}(x_{long}, x_{long}; y_w, y_l) \leq \frac{1}{3}[l_{3\eta,\frac{\gamma}{3}}(x_{long}, x_{short}; y_w, y_w) \tag{13}
$$
$$
+ l_{3\eta,\frac{\gamma}{3}}(x_{short}, x_{short}; y_w, y_l) + l_{3\eta,\frac{\gamma}{3}}(x_{short}, x_{long}; y_l, y_l)]
$$

*Proof.*

$$
\begin{aligned}
&l_{\eta,\gamma}(x_{long}, x_{long}; y_w, y_l)\\
&= f(\eta \cdot [r_\phi(x_{long}, y_w) - r_\phi(x_{long}, y_l) - \gamma])\\
&= f(\eta \cdot [r_\phi(x_{long}, y_w) - r_\phi(x_{short}, y_w)\\
&\qquad\quad + r_\phi(x_{short}, y_w) - r_\phi(x_{short}, y_w)\\
&\qquad\quad + r_\phi(x_{short}, y_w) - r_\phi(x_{long}, y_l) - \gamma])\\
&= f(\eta \cdot [\Delta_1 + \Delta_2 + \Delta_3 - \gamma])\\
&= f(\frac{1}{3}[\eta \cdot (3\Delta_1 - \gamma + 3\Delta_2 - \gamma + 3\Delta_3 - \gamma)])\\
&\leq \frac{1}{3}[f(\eta \cdot (3\Delta_1 - \gamma)) + f(\eta \cdot (3\Delta_2 - \gamma)) + f(\eta \cdot (3\Delta_3 - \gamma))] \quad \text{by Jensen's Inequality}\\
&= \frac{1}{3}[l_{3\eta,\frac{\gamma}{3}}(x_{long}, x_{short}; y_w, y_w) + l_{3\eta,\frac{\gamma}{3}}(x_{short}, x_{short}; y_w, y_l) + l_{3\eta,\frac{\gamma}{3}}(x_{short}, x_{long}; y_l, y_l)]
\end{aligned}
$$

where

$$
\begin{aligned}
\Delta_1 &= r_\phi(x_{long}, y_w) - r_\phi(x_{short}, y_w)\\
\Delta_2 &= r_\phi(x_{short}, y_w) - r_\phi(x_{short}, y_l)\\
\Delta_3 &= r_\phi(x_{short}, y_l) - r_\phi(x_{long}, y_l)
\end{aligned}
$$

$\square$

## I.2   PROOF OF THEOREM 1

*Proof.* Directly applying expectation $\mathbb{E}_{x_\cdot \sim \mathcal{D}_{x_\cdot}; y_w, y_l \sim \mathcal{D}_y \atop y_w \succ y_l | x_{long}}$ to inequality 13 and using assumption 1, we can obtain the following inequality:

$$
\begin{aligned}
&\mathcal{L}_{\eta,\gamma}(\mathcal{D}_{x_{long}}, \mathcal{D}_{x_{long}}; \mathcal{D}_{y_w \succ y_l | x_{long}}, \mathcal{D}_{y_w \succ y_l | x_{long}})\\
&\leq \frac{1}{3}
\begin{bmatrix}
\mathcal{L}_{3\eta,\frac{\gamma}{3}}(\mathcal{D}_{x_{long}}, \mathcal{D}_{x_{short}}; \mathcal{D}_{y_w | x_{short}}, \mathcal{D}_{y_w | x_{short}})\\
+ \mathcal{L}_{3\eta,\frac{\gamma}{3}}(\mathcal{D}_{x_{short}}, \mathcal{D}_{x_{short}}; \mathcal{D}_{y_w \succ y_l | x_{short}}, \mathcal{D}_{y_w \succ y_l | x_{short}})\\
+ \mathcal{L}_{3\eta,\frac{\gamma}{3}}(\mathcal{D}_{x_{short}}, \mathcal{D}_{x_{long}}; \mathcal{D}_{y_l | x_{short}}, \mathcal{D}_{y_l | x_{short}})
\end{bmatrix} . \tag{14}
\end{aligned}
$$

We prove only the second term of inequality 14, as the proofs for the remaining terms follow in the same manner.

$$\mathbb{E}_{x. \sim \mathcal{D}_{x.}; y_w, y_l \sim \mathcal{D}_y, \ y_w \succ y_l | x_{long}} [l_{3\eta, \frac{\gamma}{3}}(x_{short}, x_{short}; y_w, y_l)] \tag{15}$$

$$=\mathbb{E}_{x. \sim \mathcal{D}_{x.}; y_w, y_l \sim \mathcal{D}_y} [P(y_w \succ y_l | x_{long}) l_{3\eta, \frac{\gamma}{3}}(x_{short}, x_{short}; y_w, y_l)] \tag{16}$$

$$=\mathbb{E}_{x. \sim \mathcal{D}_{x.}; y_w, y_l \sim \mathcal{D}_y} \Big[ \frac{P(y_w \succ y_l | x_{long})}{P(y_w \succ y_l | x_{short})} P(y_w \succ y_l | x_{short}) l_{3\eta, \frac{\gamma}{3}}(x_{short}; y_w, y_l) \Big] \tag{17}$$

$$\leq \mathbb{E}_{x. \sim \mathcal{D}_{x.}; y_w, y_l \sim \mathcal{D}_y} [P(y_w \succ y_l | x_{short}) l_{3\eta, \frac{\gamma}{3}}(x_{short}; y_w, y_l)] \tag{18}$$

$$=\mathcal{L}_{3\eta, \frac{\gamma}{3}}(\mathcal{D}_{x_{short}}, \mathcal{D}_{x_{short}}; \mathcal{D}_{y_w \succ y_l | x_{short}}, \mathcal{D}_{y_w \succ y_l | x_{short}}) \tag{19}$$

Now, we prove the inequality 7. We only need to consider the sum of the first term and the third term:

$$\mathcal{L}_{3\eta, \frac{\gamma}{3}}(\mathcal{D}_{x_{long}}, \mathcal{D}_{x_{short}}; \mathcal{D}_{y_w | x_{short}}, \mathcal{D}_{y_w | x_{short}}) + \mathcal{L}_{3\eta, \frac{\gamma}{3}}(\mathcal{D}_{x_{short}}, \mathcal{D}_{x_{long}}; \mathcal{D}_{y_l | x_{short}}, \mathcal{D}_{y_l | x_{short}}) \tag{20}$$

$$=\mathbb{E}_{\substack{x. \sim \mathcal{D}_{x.}; \\ y_w, y_l \sim \mathcal{D}_y}} [P(y_w \succ y_l | x_{short})(l_{3\eta, \frac{\gamma}{3}}(x_{long}, x_{short}; y_w, y_w) + l_{3\eta, \frac{\gamma}{3}}(x_{short}, x_{long}; y_l, y_l))] \tag{21}$$

$$\leq \mathbb{E}_{\substack{x. \sim \mathcal{D}_{x.}; \\ y_w, y_l \sim \mathcal{D}_y}} [l_{3\eta, \frac{\gamma}{3}}(x_{long}, x_{short}; y_w, y_w) + l_{3\eta, \frac{\gamma}{3}}(x_{short}, x_{long}; y_l, y_l)] \tag{22}$$

$$=\mathbb{E}_{\substack{x. \sim \mathcal{D}_{x.}; \\ y_w, y_l \sim \mathcal{D}_y}} [f(\eta \cdot (3\Delta_1(y_w) - \gamma)) + f(\eta \cdot (3\Delta_3(y_l) - \gamma))] \tag{23}$$

$$=\mathbb{E}_{\substack{x. \sim \mathcal{D}_{x.}; \\ y_w, y_l \sim \mathcal{D}_y}} [f(\eta \cdot (3\Delta_1(y_w) - \gamma)) + f(\eta \cdot (3\Delta_3(y_l) - \gamma))] \tag{24}$$

$$=\mathbb{E}_{x. \sim \mathcal{D}_{x.}; y \sim \mathcal{D}_y} [f(\eta \cdot (3\Delta_1(y) - \gamma)) + f(\eta \cdot (3\Delta_3(y) - \gamma))] \tag{25}$$

$$=\mathbb{E}_{x. \sim \mathcal{D}_{x.}; y \sim \mathcal{D}_y} [f(\eta \cdot (3\Delta_1(y) - \gamma)) + f(\eta \cdot (-3\Delta_1(y) - \gamma))] \tag{26}$$

$$=\mathbb{E}_{x. \sim \mathcal{D}_{x.}; y \sim \mathcal{D}_y} [f(3\eta\Delta_1(y) - \eta\gamma)) + f(-3\eta\Delta_1(y) - \eta\gamma))] \tag{27}$$

$$\leq \mathbb{E}_{x. \sim \mathcal{D}_{x.}; y \sim \mathcal{D}_y} s(|3\eta \cdot (r_\phi(x_{short}, y) - r_\phi(x_{long}, y))|) \tag{28}$$

where $s(\cdot)$ satisfies $f(z + \gamma) + f(-z + \gamma) \leq s(|z|)$ and

$$\Delta_1(y) = r_\phi(x_{long}, y) - r_\phi(x_{short}, y) \tag{29}$$

$$\Delta_3(y) = r_\phi(x_{short}, y) - r_\phi(x_{long}, y). \tag{30}$$

$$\square$$

The introduction of $s(\cdot)$ is to unify the two terms, $r_\phi(x_{long}, y)$ and $r_\phi(x_{short}, y)$. into a single expression. This unified expression serves to **quantify the distance** between $r_\phi(x_{long}, y)$ and $r_\phi(x_{short}, y)$. Building upon this foundation, we further generalized this concept in Appendix I.5, leading to a more generalized theorem, which may provide valuable insights and inspire future research directions. Given that $s(\cdot)$ serves as an upper bound, a tighter instantiation is theoretically preferred; we provide empirical evidence for this claim in Appendix I.8.

I.3    SoLoPO FOR $f(x) = x^2$

**Proposition 1.** *Following the notation of Theorem 1, if we take $f(x) = x^2$, the inequality becomes:*

$$\mathcal{L}_{\eta, \gamma}(x_{long}) \leq \frac{1}{3}\mathcal{L}_{3\eta, \frac{\gamma}{3}}(x_{short}) + 3\eta^2 \cdot \mathbb{E}_{x. \sim \mathcal{D}_{x.}; y \sim \mathcal{D}_y} |r_\phi(x_{short}, y) - r_\phi(x_{long}, y)|^2 + \frac{2}{3}\gamma^2 \tag{31}$$

*Proof.*

$$f(x + \gamma) + f(-x + \gamma) = (x + \gamma)^2 + (-x + \gamma)^2 = 2x^2 + 2\gamma^2$$

Therefore, by using the Theorem 1, we prove this proposition.    $\square$

## I.4 SoLoPO for $f(x) = -\log \sigma(x)$

**Proposition 2.** *Following the notation of Theorem 1, if we take $f(x) = -\log \sigma(x)$, then the inequality can be:*

$$\mathcal{L}_{\eta,\gamma}(x_{long}) \leq \frac{1}{3}\mathcal{L}_{3\eta,\frac{\gamma}{3}}(x_{short}) + \eta \cdot \mathbb{E}_{x.\sim\mathcal{D}_{x.};y\sim\mathcal{D}_y}|r_\phi(x_{short},y) - r_\phi(x_{long},y)| + \frac{2}{3}\log(1 + e^{\eta \cdot \gamma}) \tag{32}$$

*Proof.*

$$\mathcal{L}_{3\eta,\frac{\gamma}{3}}(\mathcal{D}_{x_{long}}, \mathcal{D}_{x_{short}}; \mathcal{D}_{y_w|x_{short}}, \mathcal{D}_{y_w|x_{short}}) + \mathcal{L}_{3\eta,\frac{\gamma}{3}}(\mathcal{D}_{x_{short}}, \mathcal{D}_{x_{long}}; \mathcal{D}_{y_l|x_{short}}, \mathcal{D}_{y_l|x_{short}}) \tag{33}$$

$$\leq \mathbb{E}_{\substack{x.\sim\mathcal{D}_{x.};\\ y_w,y_l\sim\mathcal{D}_y}} [l_{3\eta,\frac{\gamma}{3}}(x_{long}, x_{short}; y, y) + l_{3\eta,\frac{\gamma}{3}}(x_{short}, x_{long}; y, y)] \tag{34}$$

$$\tag{35}$$

For brevity, we omit the expectation in the following derivation.

$$l_{3\eta,\frac{\gamma}{3}}(x_{long}, x_{short}; , y_w) + l_{3\eta,\frac{\gamma}{3}}(x_{short}, x_{long}; y, y) \tag{36}$$

$$= f(3\eta \cdot (r_\phi(x_{long}, y) - r_\phi(x_{short}, y)) - \eta \cdot \gamma) + f(3\eta \cdot (r_\phi(x_{short}, y) - r_\phi(x_{long}, y)) - \eta \cdot \gamma) \tag{37}$$

$$\text{we denote } r_\phi(x_{long}, y), r_\phi(x_{short}, y) \text{ as } r_l, r_s \tag{38}$$

$$= f(3\eta \cdot (r_l - r_s) - \eta \cdot \gamma) + f(3\eta \cdot (r_s - r_l) - \eta \cdot \gamma) \tag{39}$$

$$= -\log \sigma(3\eta \cdot r_l - 3\eta \cdot r_s - \eta \cdot \gamma) - \log \sigma(3\eta \cdot r_s - 3\eta \cdot r_l - \eta \cdot \gamma) \tag{40}$$

$$= -\log \frac{1}{1 + \exp\{-(3\eta \cdot r_l - 3\eta \cdot r_s - \eta \cdot \gamma)\}} - \log \frac{1}{1 + \exp\{-(3\eta \cdot r_s - 3\eta \cdot r_l - \eta \cdot \gamma)\}} \tag{41}$$

$$= -\log \frac{e^{3\eta \cdot r_l}}{e^{3\eta \cdot r_s + \eta \cdot \gamma} + e^{3\eta \cdot r_l}} - \log \frac{e^{3\eta \cdot r_s}}{e^{3\eta \cdot r_s} + e^{3\eta \cdot r_l + \eta \cdot \gamma}} \tag{42}$$

$$= -3\eta \cdot r_l - 3\eta \cdot r_s + \log(e^{3\eta \cdot r_s + \eta \cdot \gamma} + e^{3\eta \cdot r_l}) + \log(e^{3\eta \cdot r_s} + e^{3\eta \cdot r_l + \eta \cdot \gamma}) \tag{43}$$

$$\overset{(a)}{\leq} 3\mathbb{E}_{\substack{x.\sim\mathcal{D}_{x.};\\ y\sim\mathcal{D}_y}}|r_l - r_s| + 2\log(1 + e^{3\gamma}) \tag{44}$$

In the following, we prove the inequality (a).

If $r_l \leq r_s$, then

$$-3\eta \cdot r_l - 3\eta \cdot r_s + \log(e^{3\eta \cdot r_s + \eta \cdot \gamma} + e^{3\eta \cdot r_l}) + \log(e^{3\eta \cdot r_s} + e^{3\eta \cdot r_l + \eta \cdot \gamma}) \tag{45}$$

$$\leq -3\eta \cdot r_l - 3\eta \cdot r_s + \log(e^{3\eta \cdot r_s + \eta \cdot \gamma} + e^{3\eta \cdot r_s}) + \log(e^{3\eta \cdot r_s} + e^{3\eta \cdot r_s + \eta \cdot \gamma})] \tag{46}$$

$$= -3\eta \cdot r_l - 3\eta \cdot r_s + 6\eta \cdot r_s + 2\log(1 + e^{\eta \cdot \gamma})] \tag{47}$$

$$= 3\eta \cdot r_s - 3\eta \cdot r_l + 2\log(1 + e^{\eta \cdot \gamma}) \tag{48}$$

By symmetry, we can easily obtain that if $r_s \leq r_l$, then

$$-3\eta \cdot r_l - 3\eta \cdot r_s + \log(e^{3\eta \cdot r_s + \eta \cdot \gamma} + e^{3\eta \cdot r_l}) + \log(e^{3\eta \cdot r_s} + e^{3\eta \cdot r_l + \eta \cdot \gamma}) \leq 3\eta \cdot r_l - 3\eta \cdot r_s + 2\log(1 + e^{\eta \cdot \gamma}) \tag{49}$$

Therefore,

$$-3\eta \cdot r_l - 3\eta \cdot r_s + \log(e^{3\eta \cdot r_s + \eta \cdot \gamma} + e^{3\eta \cdot r_l}) + \log(e^{3\eta \cdot r_s} + e^{3\eta \cdot r_l + \eta \cdot \gamma}) \leq 3\eta \cdot |r_l - r_s| + 2\log(1 + e^{\eta \cdot \gamma}) \tag{50}$$

$\square$

## I.5 Generalization of Theorem 1

**Theorem 2.** *Let $D_p(x_1, x_2) = (\mathbb{E}_{y\sim\mathcal{D}_y}|r_\phi(x_1, y) - r_\phi(x_2, y)|^p)^{\frac{1}{p}}$. If any divergence $D(x_1, x_2)$ satisfies*

$$D_1(x_1, x_2) \leq C_1 \cdot D(x_1, x_2) \tag{51}$$

*where $C_1$ are positive constants, then the following inequality holds true for the convex function $f(x) = -\log \sigma(x)$, as in the settings of DPO, SimPO, and ORPO:*

$$\mathcal{L}_{\eta,\gamma}(\mathcal{D}_{long}, \mathcal{D}_{long}; \mathcal{D}_{y|x_{long}}, \mathcal{D}_{y|x_{long}}) \tag{52}$$

$$\leq \frac{1}{3}\mathcal{L}_{3\eta,\frac{\gamma}{3}}(\mathcal{D}_{short}, \mathcal{D}_{short}; \mathcal{D}_{y|x_{short}}, \mathcal{D}_{y|x_{short}}) + \eta \cdot C_1 \mathbb{E}_{x. \sim \mathcal{D}_{x.}}[D(x_{short}, x_{long})] + C_2 \tag{53}$$

*where $C_2 = \frac{2}{3}(\log(1 + e^{\eta \cdot \gamma}))$.*

Theorem 2 guarantees that any new distance satisfying the inequality 51 can substitute the absolute distance.

*Proof.* This theorem can be proved by directly using the proposition 2. □

### I.6 THE GENERAL FORMULA OF SoLoPO LOSS FUNCTION

The general formula of Short-to-Long Preference Optimization (SoLoPO) loss function:

$$\mathcal{L}_{SoLoPO} = \mathbb{E}_{\substack{x \sim \mathcal{D}_{x_{short}}; \\ y_w, y_l \sim \mathcal{D}_y; y_w \succ y_l}} [f(\eta \cdot [r_\phi(x, y_w) - r_\phi(x, y_l) - \gamma])]$$
$$+ \alpha \cdot \mathbb{E}_{x. \sim \mathcal{D}_{x.}; y \sim \mathcal{D}_y} s(|r_\phi(x_{short}, y) - r_\phi(x_{long}, y)|)$$

where $f$ is a convex function, and $f(x + \gamma) + f(-x + \gamma) \leq s(|x|)$ for some function $s$. $\gamma, \alpha, \eta$ are hyperparameters.

### I.7 EMPIRICAL EVIDENCE FOR ASSUMPTION 1

In this section, we verify Assumption 1:

*since $x_{long}$ contains more task-irrelevant information than $x_{short}$, making it harder for the model to distinguish preferences when conditioned on $x_{long}$:*

$$p(y_w \succ y_l | x_{long}) \leq p(y_w \succ y_l | x_{short}) \tag{54}$$

*where $p(y_w \succ y_l \mid x) = \sigma(r^*(x, y_w) - r^*(x, y_l))$, $r^*$ denotes the optimal reward model, and $\sigma$ is the sigmoid function.*

In the absence of an optimal reward model, we employ a model $\pi_{final}$ trained via DPO to estimate the reward margin for $(y_w, y_l)$, following the reward computation formulation defined in DPO:

$$r_{DPO}(x, y_w) - r_{DPO}(x, y_l) = \left(\beta \log \frac{\pi_{final}(y_w \mid x)}{\pi_{ref}(y_w \mid x)} - \beta \log \frac{\pi_{final}(y_l \mid x)}{\pi_{ref}(y_l \mid x)}\right) \tag{55}$$

Specifically, we adopt Qwen2.5-7B-Instruct as the reference policy $\pi_{ref}$, and perform **DPO training** on data with a context length of 1K to obtain the final policy $\pi_{final}$. We set the short-context length to 4K, from which we sample preference pairs $y_w \succ y_l \sim \pi_{ref}(y \mid x_{short})$, and subsequently expand them to lengths ranging from 8K to 32K to obtain $x_{long}$. **This design aims to emulate a reward model with a non-trivial scoring capacity that is nonetheless susceptible to noise, in order to meet the preconditions of our assumption**. For each length, we compute the proportion satisfying Eq. (54) based on Eq. (55).

Table 13: Proportion of cases in which Assumption 54 holds for preference pairs $(y_w \succ y_l)$ sampled from $x_{short}$ (length 4K) and evaluated on $x_{long}$ with varying lengths (8K–32K), using a model trained via DPO on a 1K context length. Longer contexts introduce additional task-irrelevant information, leading to a gradual increase in the satisfaction rate and stabilizing at approximately 95%

| Length of $x_{long}$ | 8K | 12K | 16K | 20K | 24K | 28K | 32K |
|---|---|---|---|---|---|---|---|
| **Hypothesis Validity Proportion** | 80.58% | 86.41% | 91.26% | 90.29% | 96.12% | 95.15% | 94.17% |

Results are shown in Table 13. As the context length increases, the amount of task-irrelevant information grows, and the proportion satisfying the hypothesis gradually rises, stabilizing at around $95\%$. Considering potential inaccuracies in reward estimation, such consistently high proportions provide substantial support for Assumption 1.

## I.8 The impact of the tightness of $s(\cdot)$ on SoLoPO's performance

Theoretically, a tighter $s$ corresponds to a stronger upper bound, which may improve empirical performance. In contrast, a loose upper bound often results in vague optimization directions and may cause the optimization to converge to a local optimum. For example, consider minimizing $x^4 - x^2 = x^2(x^2 - 2)$, whose global minima are at $x = 1$. If we use an upper bound $x^4$, the global minimum of this upper bound is at $x = 0$. Therefore, if the upper bound is too loose, the obtained solution may deviate from the true optimum.

We compare a *tighter* setting $s_t(x) = |x|$ (used in the paper) with a *looser* setting $s_l(x) = |x| + \sin(x) + 1 \geq s_t(x)$ on Qwen2.5-7B-Instruct, while keeping all other training configurations identical. As shown in the table below, the tighter $s(\cdot)$ generally achieves better performance, except in the MD-QA scenario of LongBenchV1 where SoLo-DPO$(s_t)$ is slightly worse than SoLo-DPO$(s_l)$. Since $s_t \leq s_l$ in our experiments, we can infer that a looser $s(\cdot)$ tends to yield worse performance.

Table 14: Impact of $s(\cdot)$ tightness on performance. Tighter $s(\cdot)$ generally improves performance.

| Model | QAs-LongBenchV1 | | | QAs-RULER | | | | |
|---|---|---|---|---|---|---|---|---|
| | S-Doc QA | M-Doc QA | Avg. | 4k | 8k | 16k | 32k | Avg. |
| $M^{DPO}_{SoLo(s_l)}$ | 35.3 | **57.8** | 46.5 | 64.2 | 62.6 | 61.2 | 57.4 | 61.4 |
| $M^{DPO}_{SoLo(s_t)}$ | **38.0** | 57.6 | **47.8** | **66.4** | **64.5** | **62.7** | **57.7** | **62.8** |
| $M^{SimPO}_{SoLo(s_l)}$ | 36.8 | 56.7 | 46.7 | 67.8 | 65.0 | 61.9 | 57.2 | 63.0 |
| $M^{SimPO}_{SoLo(s_t)}$ | **38.1** | **58.6** | **48.4** | **69.2** | **66.0** | **62.7** | **57.8** | **63.9** |
| $M^{ORPO}_{SoLo(s_l)}$ | 36.1 | 57.8 | 46.9 | 68.1 | 65.4 | 60.5 | 56.2 | 62.6 |
| $M^{ORPO}_{SoLo(s_t)}$ | **37.6** | **61.4** | **49.5** | **70.8** | **68.3** | **64.0** | **57.3** | **65.1** |

## I.9 Discussion on the Modeling of SoLoPO

**a. Two key abilities in long-context scenarios**  Unlike short-context tasks such as mathematics [39], which can directly leverage the model's inherent **(contextual knowledge) reasoning** ability, long-context tasks—such as question answering [64] or information extraction [73]—also require the model to possess **contextual knowledge localization** skills, i.e., the ability to identify task-relevant information $c_{rel}$ from a long context $c_{long}$ while ignoring irrelevant content $c_{irr}$. In other words, the model needs to first identify the key task-relevant information $c_{rel}$ from the context $c_{long}$ and subsequently perform reasoning upon it [38].

**b. Explicit modeling of two key abilities in SoLoPO**  Recall that the optimization objective of SoLoPO is defined as follows:

$$\mathcal{L}_{SoLoPO} = \mathbb{E}_{\substack{x \sim \mathcal{D}_{x_{short}}; y_w, y_l \sim \mathcal{D}_y \\ y_w \succ y_l \sim \pi_\theta(y|x_{short})}} \Big[ f\Big(3\eta \cdot [r_\phi(x, y_w) - r_\phi(x, y_l) - \frac{\gamma}{3}]\Big)\Big] \underbrace{\phantom{xxxxxxxxxxxxxxxxxxxxxxxxxxxx}}_{\text{short-context preference optimization}} \tag{56}$$

$$+ \alpha \cdot \mathbb{E}_{\substack{x. \sim \mathcal{D}_{x.}; y. \sim \mathcal{D}_{(y_w, y_l)} \\ y_w \succ y_l \sim \pi_\theta(y|x_{short})}} \underbrace{[s(3\eta \cdot |r_\phi(x_{short}, y) - r_\phi(x_{long}, y)|)]}_{\text{short-to-long reward alignment}}. \tag{57}$$

SoLoPO explicitly models the two abilities in a decoupled manner:

- **Contextual knowledge reasoning:** Since $x_{short}$ consists of the task instruction $I$ and the task-relevant content $c_{rel}$, i.e., $x_{short} := [c_{rel}; I]$, SoLoPO directly enhances the model's inherent reasoning ability via short-context preference optimization (Eq. (56)).

- **Contextual knowledge localization:** SoLo-RA (Eq. 57) encourages the reward model $r_\phi$ to implicitly predict $\hat{x}_{short} \sim \hat{p}(x_{short} \mid x_{long})$ by minimizing the divergence between $\hat{x}_{short} := [\hat{c}_{rel}; I]$ and the ground-truth $x_{short} := [c_{rel}; I]$. In preference optimization, where the reward model $r_\phi$ and the policy model $\pi_\theta$ coincide, this process also strengthens the policy model's ability to locate relevant knowledge $c_{rel}$ for task $I$ within a long context $c_{long}$.

  Taking SimPO as an example, where $s(|x|) = |x| + C$ and the reward is defined as $r_\theta(x, y) = \frac{\beta}{|y|} \log \pi_\theta(y|x)$. For brevity, we set $\eta = \frac{1}{3}$ and omit constant $C$, the SoLo-RA loss becomes:

  $$\text{SoLo-RA}_{SimPO} = \mathbb{E}_{\substack{x_. \sim \mathcal{D}_{x_.}; y \sim \mathcal{D}_y \\ y \sim \pi_\theta(y|x_{short})}} \left[ \frac{\beta}{|y|} |\log \pi_\theta(y \mid x_{short}) - \log \pi_\theta(y \mid x_{long})| \right], \quad (58)$$

  which encourages $\pi_\theta$ to produce an output $y$ with the same likelihood whether it is conditioned on $x_{short}$ or on the full input $x_{long}$. Under this objective, SoLo-RA **implicitly** guides the model to extract from $x_{long}$ only the minimal sufficient information needed to behave as if conditioned on $x_{short}$. That is, it enforces:

  $$\pi_\theta(y \mid x_{long}) \approx \pi_\theta(y \mid x_{short}) \overset{implicitly}{\Longrightarrow} g_{\theta'}(x_{long}) = x_{short} \quad (59)$$

  where $g_{\theta'}$ can be interpreted as an internal attention mechanism or latent projection that compresses $x_{long}$ into a representation functionally equivalent to $x_{short}$.

  This learned behavior aligns with the principle of contextual knowledge localization.

### I.10 SUPPORTING ANALYSIS FOR CHOSEN-ONLY SOLO-RA

#### I.10.1 THE ORIGINAL OBJECTIVE OF SOLOPO IS THEORETICAL AND EMPIRICAL SUPPORTED

From an theoretical perspective, SoLoPO decomposes long-context preference optimization (PO) into short-context PO and short-to-long reward alignment (SoLo-RA) (§ 2.2). The experimental analysis in Section 4.2 and Figure 2a shows that directly using the SoLoPO objective—applying SoLo-RA jointly to both the chosen and rejected responses (*both* SoLo-RA)—achieves superior performance to Long-PO across most settings. These results offer both theoretical proof and empirical validation for the original optimization objective of **SoLoPO with *both* SoLo-RA**, as defined in Eq. (57).

#### I.10.2 SUPPORTING ANALYSIS FOR CHOSEN-ONLY SOLO-RA

In practical scenarios, we further consider a variant, **SoLoPO with *chosen-only* SoLo-RA**, which is motivated by two factors:

1. $y_l$ may not always exploit the key information in the context. For example, $y_l$ might simply respond, "No task-relevant content can be found in the context, so the question cannot be answered." In such cases, enforcing SoLo-RA may not improve, and could even degrade, the model's ability to locate or reason over relevant long-context information.

2. We aim to reduce the amount of long-text processing during training, thereby improving training efficiency.

In addition, we further examine the theoretical validity of the first motivation from a **data-sampling perspective**.

**a. Relation $\pi(y \mid x_{short}) \geq \pi(y \mid x_{long})$ typically holds owing to the practical data sampling strategy.** Recall that preference pairs in SOLOPO are obtained based on short contexts, as defined in Eq. (57):

$$y_w \succ y_l \sim \pi_\theta(y|x_{short}). \quad (60)$$

For arbitrary $x_{short}$ and $x_{long}$, the Kullback–Leibler divergence [68] satisfies:

$$D_{KL}(\pi(y|x_{short}) \parallel \pi(y|x_{long})) = \int \left[ \log \pi(y|x_{short}) - \log \pi(y|x_{long}) \right] \pi(y|x_{short}) \, dy \geq 0. \quad (61)$$

This implies that, when sampling $y \sim \pi_\theta(y \mid x_{short})$, the quantity $\log \pi(y \mid x_{short}) - \log \pi(y \mid x_{long})$ is more likely than not to be non-negative. Since the logarithm is strictly monotonic, the following inequality **tends to hold** (more accurately, holds in expectation; see Table 15 for empirical evidence):

$$\pi_\theta(y \mid x_{short}) \geq \pi_\theta(y \mid x_{long}), \quad \text{for} \quad y \sim \pi_\theta(y|x_{short}). \quad (62)$$

**b. Applying SoLo-RA to $y_l$ may harm long-context capability.** Referring to Table 1, the SoLo-RA loss becomes:

$$\text{SoLo-RA}_{SimPO} = \mathbb{E}_{\substack{x. \sim \mathcal{D}_{x.}; y \sim \mathcal{D}_y \\ y \sim \pi_\theta(y|x_{short})}} \left[ \frac{\beta}{|y|} \big| \log \frac{\pi_\theta(y|x_{short})}{\pi_\theta(y|x_{long})} \big| \right] \tag{63}$$

$$\text{SoLo-RA}_{DPO} = \mathbb{E}_{\substack{x. \sim \mathcal{D}_{x.}; y \sim \mathcal{D}_y \\ y \sim \pi_\theta(y|x_{short})}} \left[ \beta \big| \log \frac{\pi_\theta(y|x_{short})}{\pi_\theta(y|x_{long})} + \log \frac{\pi_{ref}(y|x_{long})}{\pi_{ref}(y|x_{short})} + \log \frac{Z(x_{short})}{Z(x_{long})} \big| \right] \tag{64}$$

$$\text{SoLo-RA}_{ORPO} = \mathbb{E}_{\substack{x. \sim \mathcal{D}_{x.}; y \sim \mathcal{D}_y \\ y \sim \pi_\theta(y|x_{short})}} \left[ \big| \log \frac{\pi_\theta(y|x_{short})}{\pi_\theta(y|x_{long})} + \log \frac{1 - \pi_\theta(y|x_{long})}{1 - \pi_\theta(y|x_{short})} \big| \right] \tag{65}$$

Since the latter two terms in SoLo-RA$_{DPO}$ are independent of the learnable parameters $\theta$, and the reward coefficient does not affect the subsequent analysis, we treat the SoLo-RA in DPO as equivalent to that in SimPO. Expanding the expectation, we separate **cases** based on the likelihood ratio:

$$\text{SoLo-RA}_{DPO/SimPO} = \mathbb{E}_{\substack{x. \sim \mathcal{D}_{x.}; y \sim \mathcal{D}_y \\ y \sim \pi_\theta(y|x_{short})}} \left[ \mathbf{1}\big[ \frac{\pi_\theta(y|x_{short})}{\pi_\theta(y|x_{long})} \geq 1 \big] \log \frac{\pi_\theta(y|x_{short})}{\pi_\theta(y|x_{long})} \right] \tag{66}$$

$$- \mathbb{E}_{\substack{x. \sim \mathcal{D}_{x.}; y \sim \mathcal{D}_y \\ y \sim \pi_\theta(y|x_{short})}} \left[ \mathbf{1}\big[ \frac{\pi_\theta(y|x_{short})}{\pi_\theta(y|x_{long})} < 1 \big] \log \frac{\pi_\theta(y|x_{short})}{\pi_\theta(y|x_{long})} \right] \tag{67}$$

$$\text{SoLo-RA}_{ORPO} = \mathbb{E}_{\substack{x. \sim \mathcal{D}_{x.}; y \sim \mathcal{D}_y \\ y \sim \pi_\theta(y|x_{short})}} \left[ \mathbf{1}\big[ \frac{\pi_\theta(y|x_{short})}{\pi_\theta(y|x_{long})} \geq 1 \big] (\log \frac{\pi_\theta(y|x_{short})}{\pi_\theta(y|x_{long})} + \log \frac{1 - \pi_\theta(y|x_{long})}{1 - \pi_\theta(y|x_{short})}) \right] \tag{68}$$

$$- \mathbb{E}_{\substack{x. \sim \mathcal{D}_{x.}; y \sim \mathcal{D}_y \\ y \sim \pi_\theta(y|x_{short})}} \left[ \mathbf{1}\big[ \frac{\pi_\theta(y|x_{short})}{\pi_\theta(y|x_{long})} < 1 \big] (\log \frac{\pi_\theta(y|x_{short})}{\pi_\theta(y|x_{long})} + \log \frac{1 - \pi_\theta(y|x_{long})}{1 - \pi_\theta(y|x_{short})}) \right] \tag{69}$$

We restrict our analysis to the cases in SoLo-RA$_{\text{DPO/SimPO}}$, as SoLo-RA$_{\text{ORPO}}$ can be examined in the same manner. We next consider two cases:

- **Case 1: Winning Responses** ($y = y_w$) Based on Eq. (62), we have $\pi_\theta(y_w|x_{long}) \leq \pi_\theta(y_w|x_{short})$. Here, SoLo-RA minimizes $\log \frac{\pi_\theta(y_w|x_{short})}{\pi_\theta(y_w|x_{long})}$, which:
    - **Decrease** $\pi_\theta(y_w|x_{short})$ (nominator) but is counterbalanced by first loss term (short-context preference optimization);
    - **Increases** $\pi_\theta(y_w|x_{long})$ (denominator), aligning the objective of long-context alignment.

  Since SoLoPO also includes short-context PO (Eq. (56)), this term dominates and can prevent $\pi_\theta(y_w|x_{short})$ from decreasing, thereby mitigating the negative impact on the model's short-context performance.

- **Case 2: Losing Responses** ($y = y_l$) Based on Eq. (62), we have $\pi_\theta(y_l|x_{long}) \leq \pi_\theta(y_l|x_{short})$. SoLo-RA again minimizes $\log \frac{\pi_\theta(y_l|x_{short})}{\pi_\theta(y_l|x_{long})}$, leading to:
    - **Decrease** in $\pi_\theta(y_l|x_{short})$ (desirable for reducing poor generation)
    - **Increase** in $\pi_\theta(y_l|x_{long})$ (undesirable, as it promotes $y_l$ in long contexts).

  However, there is no other loss here that can reduce the prediction probability of $\pi_\theta(y_l|x_{long})$, therefore, using **Case 2** would bring certain negative impacts to the model's long-text capabilities.

Based on the above analysis and our goal of improving training efficiency, we adopt the *chosen-only* SoLo-RA in practical applications of SoLoPO. The experimental results in Section 4.2 and Figure 2a further demonstrate the effectiveness of the *chosen-only* SoLo-RA.

**c. No conflict with the original objective of SoLoPO** It is important to note that **the above analysis does not conflict with the original optimization objective of SOLOPO (with *both* SoLo-RA)**. As long as the objective is perfectly optimized—e.g., the SoLo-RA loss (Eq. (57)) reaches zero—the short-context PO (Eq. (56)) will simultaneously reduce $\pi_\theta(y_l|x_{short})$ and $\pi_\theta(y_l|x_{long})$, thereby resolving the issue in **Case 2** and aligning with the objective of preference optimization.

**d. Empirical validation of relationship** $\pi_\theta(y \mid x_{\textbf{short}}) \geq \pi_\theta(y \mid x_{\textbf{long}})$   We further empirically validate the relationship $\pi_\theta(y \mid x_{\text{short}}) \geq \pi_\theta(y \mid x_{\text{long}})$ for $y \sim \pi_\theta(y|x_{short})$ (Eq. (62)). Specifically, 100 preference pairs are sampled from $x_{\text{short}}$ (length 1K) using Qwen-2.5-7B-Instruct, and the corresponding contexts are then extended to lengths from 4K to 32K in 4K increments to form $x_{\text{long}}$. For each length, we measure the proportion of instances in which the relationship holds. As shown in Table 15, the relationship is preserved in $100\%$ of the cases across all context lengths tested.

Table 15: Proportion of cases where relationship $\pi(y \mid x_{\text{short}}) \geq \pi(y \mid x_{\text{long}})$ holds for preference pairs $(y_w \succ y_l)$ sampled from $x_{\text{short}}$ (length 1K) evaluated on $x_{\text{long}}$ with varying lengths.

| Length of $x_{long}$ | 4K | 8K | 12K | 16K | 20K | 24K | 28K | 32K |
|---|---|---|---|---|---|---|---|---|
| **Ratio of** $\pi_\theta(y_w \mid x_{\textbf{short}}) \geq \pi_\theta(y_w \mid x_{\textbf{long}})$ | 100% | 100% | 100% | 100% | 100% | 100% | 100% | 100% |
| **Ratio of** $\pi_\theta(y_l \mid x_{\textbf{short}}) \geq \pi_\theta(y_l \mid x_{\textbf{long}})$ | 100% | 100% | 100% | 100% | 100% | 100% | 100% | 100% |

## I.11  SUPPORTING ANALYSIS OF SOLOPO'S STABILITY IN SHORT-CONTEXT CAPABILITY

Appendix I.10.2 analyzes the rationale of the *chosen-only* SoLo-RA. In **Case 1**, it is noted that during optimization, $\pi_\theta(y_w|x_{short})$ may decrease. However, since SoLoPO also incorporates short-context PO (Eq. (56)), this term dominates and prevents $\pi_\theta(y_w|x_{short})$ from dropping, thereby alleviating distributional shift in short contexts [16] and mitigating its adverse impact on short-context performance.

## I.12  APPLICATIONS OF SHORT-TO-LONG PREFERENCE OPTIMIZATION LOSS

Table 16: Variants of Convex function $f(x)$ and upper bound function $s(x)$.

| **Methods** | $f(x)$ | $s(x)$ | $r_\phi(x,y)$ |
|---|---|---|---|
| DPO[54] | $\log(1 + e^{-x})$ | $\|x\| + 2\log(1 + e^{3\gamma})$ | $\beta \log \frac{\pi_\theta(y|x)}{\pi_{ref}(y|x)} + \beta \log Z(x)$ |
| SimPO[48] | $\log(1 + e^{-x})$ | $\|x\| + 2\log(1 + e^{3\gamma})$ | $\frac{\beta}{\|y\|} \log \pi_\theta(y|x)$ |
| ORPO[30] | $\log(1 + e^{-x})$ | $\|x\| + 2\log(1 + e^{3\gamma})$ | $\log \frac{\pi_\theta(y|x)}{1-\pi_\theta(y|x)}$ |
| IPO [1] | $x^2$ | $2x^2 + 2\gamma^2$ | $\log \frac{\pi_\theta(y|x)}{\pi_{ref}(y|x)}$ |
| SLiC [85] | $max(0, -x)$ | $\|x\| - \gamma$ | $\log \pi_\theta(y|x)$ |
| - | $e^{-x}$ | $2e^{-\gamma}cosh(\|x\|)$ | - |
| - | $(max(0, -x))^2$ | $x^2 - \gamma^2$ | - |

In Section 2.3, we apply SOLOPO to several mainstream preference optimization (PO) algorithms, including DPO, SimPO, and ORPO. More generally, SOLOPO is compatible with any PO algorithm for which there exists an upper-bound function $s(x)$ of its convergence function $f(x)$ such that

$$f(x + \gamma) + f(-x + \gamma) \leq s(x). \tag{70}$$

Table 16 lists each PO algorithm with its corresponding $f(x)$ and $s(x)$. It is worth noting that if the inequality is tight, the performance would be further enhanced theoretically. If $s(x)$ is perfectly tight, *i.e.* $f(x + \gamma) + f(-x + \gamma) = s(x)$, then $s(x)$ should satisfy the following properties:

- $s(x)$ is an even function
- $\forall x, \ s(x) \geq 2 \cdot f(\gamma) = s(0)$

We subsequently present the complete theoretical objectives for all considered PO algorithms.

**DPO Setting:**

$$\mathcal{L}_{SoLoPO}^{DPO} = -\mathbb{E}_{\substack{x\sim\mathcal{D}_{x_{short}}; \\ y_w,y_l\sim\mathcal{D}_y; y_w\succ y_l}} [\log\sigma(3\cdot[\beta\log\frac{\pi_\theta(y_w|x)}{\pi_{ref}(y_w|x)} - \beta\log\frac{\pi_\theta(y_l|x)}{\pi_{ref}(y_l|x)} - \gamma])]$$

$$+ 3\cdot\mathbb{E}_{x.\sim\mathcal{D}_{x.};y\sim\mathcal{D}_y}|\beta\log\frac{\pi_\theta(y|x_{short})}{\pi_{ref}(y|x_{short})} - \beta\log\frac{\pi_\theta(y|x_{long})}{\pi_{ref}(y|x_{long})}|$$

**SimPO Setting:**

$$\mathcal{L}_{SoLoPO}^{SimPO} = -\mathbb{E}_{\substack{x \sim \mathcal{D}_{x_{short}}; \\ y_w, y_l \sim \mathcal{D}_y; y_w \succ y_l}} [\log \sigma(3 \cdot [\frac{\beta}{|y_w|} \log \pi_\theta(y_w|x) - \frac{\beta}{|y_l|} \log \pi_\theta(y_l|x) - \gamma])]$$

$$+ 3 \cdot \mathbb{E}_{x. \sim \mathcal{D}_{x.}; y \sim \mathcal{D}_y} |\frac{\beta}{|y|} \log \pi_\theta(y|x_{short}) - \frac{\beta}{|y|} \log \pi_\theta(y|x_{long})|$$

**ORPO Setting:**

To maintain consistency with the vanilla ORPO, we add the conventional causal language modeling negative log-likelihood (NLL) loss.

$$\mathcal{L}_{SoLoPO}^{ORPO} = -\mathbb{E}_{\substack{x \sim \mathcal{D}_{x_{short}}; \\ y_w, y_l \sim \mathcal{D}_y; y_w \succ y_l}} [\log \sigma(3 \cdot [\log \frac{\pi_\theta(y_w|x)}{1 - \pi_\theta(y_w|x)} - \log \frac{\pi_\theta(y_l|x)}{1 - \pi_\theta(y_l|x)} - \gamma])]$$

$$+ 3 \cdot \mathbb{E}_{x. \sim \mathcal{D}_{x.}; y \sim \mathcal{D}_y} |\log \frac{\pi_\theta(y|x_{short})}{1 - \pi_\theta(y|x_{short})} - \log \frac{\pi_\theta(y|x_{long})}{1 - \pi_\theta(y|x_{long})}|$$

$$+ \mathbb{E}_{\substack{x \sim \mathcal{D}_{x_{short}}; \\ y_w \sim \mathcal{D}_y}} \mathcal{L}_{NLL}(\pi_\theta; x; y_w)$$

**IPO Setting:**

$$\mathcal{L}_{SoLoPO}^{IPO} = \mathbb{E}_{\substack{x \sim \mathcal{D}_{x_{short}}; \\ y_w, y_l \sim \mathcal{D}_y; y_w \succ y_l}} [3 \cdot [\log \frac{\pi_\theta(y_w|x)}{\pi_{ref}(y_w|x)} - \log \frac{\pi_\theta(y_l|x)}{\pi_{ref}(y_l|x)} - \gamma]]^2$$

$$+ 18 \cdot \mathbb{E}_{x. \sim \mathcal{D}_{x.}; y \sim \mathcal{D}_y} [\log \frac{\pi_\theta(y|x_{short})}{\pi_{ref}(y|x_{short})} - \log \frac{\pi_\theta(y|x_{long})}{\pi_{ref}(y|x_{long})}]^2$$

**SLiC Setting:**

$$\mathcal{L}_{SoLoPO}^{SLiC} = \mathbb{E}_{\substack{x \sim \mathcal{D}_{x_{short}}; \\ y_w, y_l \sim \mathcal{D}_y; y_w \succ y_l}} [\max(0, -\log \pi_\theta(y_w|x) + \log \pi_\theta(y_l|x) + \gamma)]$$

$$+ \mathbb{E}_{x. \sim \mathcal{D}_{x.}; y \sim \mathcal{D}_y} |\log \pi_\theta(y|x_{short}) - \log \pi_\theta(y|x_{long})|$$

