# OpenReview forum: "SoLoPO: Unlocking Long-Context Capabilities in LLMs via Short-to-Long Preference Optimization"
_ICLR.cc/2026/Conference — ICLR 2026 Poster_

### Official Review · Reviewer_QMaK · 2025-10-31

**Soundness:** 3
**Presentation:** 3
**Contribution:** 3
**Rating:** 6
**Confidence:** 4

**Summary:**

This paper introduces SoLoPO (Short-to-Long Preference Optimization), a framework designed to improve the long-context capabilities of large language models (LLMs) by decoupling long-context preference optimization into two components:
1. Short-context PO: Enhances reasoning on compressed, task-relevant contexts.
2. Short-to-Long Reward Alignment (SoLo-RA): Encourages consistency in reward scores between short and long contexts containing the same task-relevant information.

The authors provide theoretical justification for this decoupling and empirically validate SoLoPO across multiple preference optimization algorithms (DPO, SimPO, ORPO) and models (Qwen2.5, Llama3.1). Key results show that SoLoPO improves performance on long-context benchmarks (e.g., LongBench, RULER, NIAH-Plus) while maintaining short-context capabilities and significantly boosting training efficiency.

**Strengths:**

1. A theoretical decomposition of long-context PO into short-context PO and SoLo-RA.
2. The idea of decoupling long-context alignment into short-context reasoning and cross-context reward alignment is novel and well-motivated.
3. Offers a practical and scalable solution with clear efficiency gains (e.g., 2.1× longer trainable sequences, 52% runtime reduction).

**Weaknesses:**

1. The theory relies on the redundancy hypothesis and Assumption 1, which, while empirically supported, may not hold for all long-context tasks (e.g., when all context is relevant).
2. The synthetic dataset construction (mixing relevant and irrelevant documents) is simple but may not reflect real-world long-context complexity.

**Questions:**

1. How does SoLoPO perform on non-QA long-context tasks such as summarization, multi-turn dialogue, or document-level translation?
2. Can the framework be extended to generation-heavy tasks where both input and output are long? The current focus is on long-input scenarios.

---

> ### Author Response · Authors · 2025-11-20
> **Response to Reviewer QMaK**
>
> We sincerely thank the reviewer for the thoughtful and positive assessment. We are pleased that you recognize the novelty and strong motivation behind our theoretical decomposition of long-context PO into short-context PO and SoLo-RA. Your acknowledgment of our approach as a practical and scalable solution, highlighted by the clear efficiency gains is greatly appreciated. Based on your insightful feedback, we have provided the following responses:
>
> > **Weakness1:Regarding the concern that the redundancy hypothesis and Assumption 1 may not be applicable to all long-context tasks (e.g., when all context is relevant)**:
>
> Thank you for highlighting this detail. As discussed in our "Limitations & Future Work" section (Appendix C):
> - **SoLoPO is designed to model the ability of contextual knowledge localization and utilization within long context scenarios**. As discussed in Lines 201-202, when all contexts are relevant, short and long texts become equivalent. In such scenarios, SoLoPO degenerates into standard preference optimization, as the focus shifts to modeling the model's ability to utilize contextual knowledge rather than its capacity for contextual knowledge localization, since all contextual information are task-relevant.
> - **Potential Approach Based on Intrinsic Model Redundancy**: Standard attention within LLMs inherently exhibits redundancy[1,2,3]. This intrinsic redundancy underpins techniques such as KV compression[3] and sparse attention[1]. We hypothesize that this model-inherent redundancy can be leveraged for representational compression (e.g. gist-token[2]/KV compression[3]). Consequently, for tasks lacking linguistic redundancy, the Short2Long paradigm could potentially be applied based on the model-inherent redundancy, thereby reducing training time and memory consumption for long-text scenarios.
>
> > **Weakness2: Regarding the concern that a simple mixture of relevant and irrelevant texts might not capture real-world long-context complexities.**
>
> Thank you for this insightful comment.
> - As discussed in our Related Work section, current approaches to long-context alignment primarily focus on two angles: data augmentation and training objective optimization. Our paper predominantly investigates the latter. We theoretically derive an optimization objective tailored for long-context scenarios, which simultaneously enhances both data construction and training efficiency.
> - Furthermore, this straightforward data synthesis approach possesses generalization capabilities. Experiments on real-world long-context benchmarks, such as LongbenchV1-QAs (Table 1) and LongBenchV2 (Table 4), show that SoLoPO outperforms the original PO algorithm.
> - Besides, we also agree that high-quality long-context data augmentation is crucial. We believe that combining SoLoPO with meticulously constructed high-quality or real-world data will yield superior result. This represents a highly promising research avenue, as discussed in our "Limitations & Future Work" section (Appendix C).
> ```
> [1] SeerAttention: Self-distilled Attention Gating for Efficient Long-context Prefilling (NeurIPS 2025)
> [2] Long Context Compression with Activation Beacon (ICLR 2026)
> [3] SnapKV: LLM Knows What You are Looking for Before Generation (NeurIPS 2024)
> ```

---

> > ### Author Response · Authors · 2025-11-20
> > **Response to Reviewer QMaK**
> >
> > > **Question1: Regarding how SoLoPO performs on non-QA tasks.**
> >
> > Thank you for the valuable feedback.
> > - Our primary focus is on long-context understanding tasks. Existing researches[6,7] highlight two critical capabilities for long-context scenarios, i.e. contextual knowledge localization and reasoning. In addition, long-context QA tasks have predominantly employed to evaluate these two abilities in prior works[8,9,11]. Thus, We follow the evaluation setting.
> >
> > - Furthermore, contextual inputs that are completely task-relevant in the suggested tasks such as long-context summarization and translation. Under such conditions, SoLoPO degenerates into standard preference optimization, as we mentioned in Line 200. Moreover, these tasks also necessitate strong generation capabilities of models, which falls outside the scope of our current investigation.
> >
> > - Following your valuble suggestion, we conduct additional evaluations on summarization and multi-turn dialogue tasks using the model trained solely on QA data. Specifically, under the greedy decoding setting, we evaluate performance in:
> >     - **Summarization tasks** in LongBenchV1[5] (Sum), evaluated by RougeLsum.
> >     - **Dialogue understanding task** in ∞BENCH[4] (Dia), evaluated by Sub-EM.
> >
> >     |Model|Sum|||||Dia|
> >     |---|---|---|---|---|---|---|
> >     ||gov_report|qmsum|multi_news|vcsum|**Avg. Sum**||
> >     |**Qwen2.5-7B**|
> >     |Instruct|35.2|23.5|24.8|15.9|24.9|14.56|
> >     ||
> >     |$M^{\text{DPO}}_{\text{Short}}$|35.1|23.7|24.6|16.0|24.9|12.62|
> >     |$M^{\text{DPO}}_{\text{Long}}$|35.0|23.5|24.5|16.1|24.8|12.62|
> >     |$M^{\text{DPO}}_{\text{SoLo}}$|34.4|23.4|24.8|16.2|24.7|14.00|
> >     ||
> >     |$M^{\text{SimPO}}_{\text{Short}}$|**35.5**|23.7|24.8|16.1|25.0|13.59|
> >     |$M^{\text{SimPO}}_{\text{Long}}$|35.2|23.6|**24.9**|16.1|24.9|14.56|
> >     |$M^{\text{SimPO}}_{\text{SoLo}}$|35.1|23.8|**24.9**|16.1|25.0|**15.53**|
> >     ||
> >     |$M^{\text{ORPO}}_{\text{Short}}$|35.1|23.7|**24.9**|16.1|25.0|12.62|
> >     |$M^{\text{ORPO}}_{\text{Long}}$|**35.5**|23.5|24.8|**16.4**|25.0|14.56|
> >     |$M^{\text{ORPO}}_{\text{SoLo}}$|35.1|**24.2**|24.6|**16.4**|**25.1**|14.56|
> >     ||
> >     |**Llama-3.1-8B**|
> >     |Instruct|35.2|25.3|27.0|16.8|26.1|19.42|
> >     ||
> >     |$M^{\text{ORPO}}_{\text{Short}}$|**35.8**|25.0|27.1|17.2|26.2|19.41|
> >     |$M^{\text{ORPO}}_{\text{Long}}$|35.2|25.2|**27.4**|16.3|26.0|**20.39**|
> >     |$M^{\text{ORPO}}_{\text{SoLo}}$|35.7|**25.5**|27.2|**17.3**|**26.4**|**20.39**|
> >     - **Results and Analysis**: on various PO algorithms and backbone models, models trained solely on QA data exhibit performance comparable to instruct models on both tasks, without showing significant degradation.
> >         - **Regarding the summarization task**:
> >             - Different PO algorithms perform comparably to instruct models, suggesting while these PO algorithms effectively enhance the model's capabilities on QA tasks, they do not compromise its long context capabilities related to summarization. This observation may indicate that the capabilities necessitated by summarization differ from those required for QA tasks.
> >         - **Regarding the dialogue task**:
> >             - SoLoPO demonstrates comparable or superior performance compared to the Long-PO algorithm. This could be attributed to SoLoPO's decoupling approach, which may better enhance the model's ability to locate task-critical information within the context. Additionally, SoLoPO offers significant training efficiency gains over Long-PO.
> > - Recognizing the critical importance of long-text generation, we have dicussed theoretical extensions and data augmentation strategies to enhance performance and efficiency in these scenarios, detailed in the "Limitation & Future Work" section (Appendix C) and our response to Question 2.
> >
> > ```
> > [4] ∞BENCH: Extending Long Context Evaluation
> > Beyond 100K Tokens (ACL 2024)
> > [5] LongBench: A Bilingual, Multitask Benchmark for Long Context Understanding (ACL 2024)
> > [6] Fundamental capabilities and applications of large language models: A survey (ACM Computing Surveys)
> > [7] A Comprehensive Survey on Long Context Language Modeling
> > [8] ♫ MuSiQue: Multihop Questions via Single-hop Question Composition (TACL 2022)
> > [9] Making Long-Context Language Models Better Multi-Hop Reasoners (ACL 2024)
> > ```

---

> ### Author Response · Authors · 2025-11-20
> **Response to Reviewer QMaK**
>
> > **Question2: Regarding SoLoPO's current focus on long-context input and its potential application to generation-heavy tasks.**
>
> We greatly appreciate your observation and agree this is an important area for future exploration.
> - As mentioned in our limitations, SoLoPO currently focuses primarily on long-context input understanding and has not extended the theoretical framework and experiments to long-context generation. The main reason is that long-context understanding[11] and long text generation[10] present fundamentally different challenges, and most existing work addresses them separately.
>
> - **Potential Extension**: Our framework's core idea is to establish a proper connection between long and short inputs: as long as the essential task-critical information is preserved, the task can be performed equivalently regardless of the input text's length. We believe this concept of equivalence can be analogously applied to long-output generation tasks. For instance:
>     - **Long Chain-of-Thought (CoT) Tasks**: Similar to our approach, whether the CoT is short or long (potentially including more trial-and-error or reflection), if the final output is correct and logically sound, the two can be considered equivalent from a task completion perspective.
>     - **Text Refinement Tasks**: A more polished text, often longer due to richer rhetorical devices, can be considered semantically and logically equivalent to a plainer, shorter version, provided the core meaning is retained.
>     - **Story Generation Tasks**: Longer story outputs might offer more detailed descriptions compared to shorter ones. However, if a user specifies a narrative arc or a core plot, the shorter and longer versions can be considered equivalent in terms of fulfilling that overarching story structure.
>
> We sincerely appreciate your valuable feedback, which has helped us recognize the potential to generalize our theory and methodology to broader scenarios. In the revised version, we will add a Discussion section to explore the applicability of the SoLoPO framework to other tasks and its potential extensions.
>
> ```
> [10] LongDPO: Unlock Better Long-form Generation Abilities for LLMs via
> Critique-augmented Stepwise Information (ACL 2025)
> [11] Self-Taught Agentic Long-Context Understanding (ACL 2025)
> ```

---

> ### Comment · Reviewer_QMaK · 2025-11-26
>
> Thank you for your response. I agree that the primary focus of this paper is on long-context understanding tasks, and I will maintain my positive score.

---

> > ### Author Response · Authors · 2025-11-27
> > **Appreciation for Your Constructive Review**
> >
> > Dear Reviewer QMaK,
> >
> > Thank you very much for the time and effort you have invested in reviewing our work, as well as for your constructive feedback and positive attitude toward our research.
> >
> > Following your suggestion, we have expanded Appendix C (Limitations & Future Work) to include a deeper discussion on SoLoPO's potential for long-text generation and for accelerating fully context-relevant tasks by leveraging the intrinsic redundancy of LLMs.
> >
> > We thank you again for your thorough evaluation and valuable insights.
> >
> > Best regards,
> >
> > Authors

---

### Official Review · Reviewer_mA21 · 2025-10-31

**Soundness:** 3
**Presentation:** 2
**Contribution:** 4
**Rating:** 4
**Confidence:** 3

**Summary:**

This paper addresses the challenge of long-context alignment and aims to improve context utilization in large language models. The authors propose a theoretically grounded framework that leverages short-context preference optimization to enhance performance in long-context settings. The key idea is to decouple long-context preference optimization into two components: short-context preference optimization and short-to-long reward alignment. This framework, termed SoLoPO, can be applied to existing preference optimization methods such as DPO, SimPO, and ORPO. Experimental results across multiple benchmarks demonstrate the effectiveness of the proposed approach.

**Strengths:**

1. **Novel framework.** Proposes the SoLoPO framework to transfer short-context preference optimization capabilities to long-context alignment.
2. **Theoretical foundation.** The method is supported by solid theoretical results that justify the proposed decoupling.
3. **General applicability.** The framework can be integrated with multiple preference alignment methods, showing consistent improvements across them.
4. **Strong long-context performance.** The chosen-only SoLoPO variant consistently outperforms standard PO baselines on long-context benchmarks.

**Weaknesses:**

1. **Degraded short-context performance.** The method shows reduced performance on the short-context Open LLM Leaderboard. Lines 103 and 377 claim that SoLoPO maintains short-context performance, yet Table 4 indicates otherwise. SoLoPO underperforms the PO baseline in 16 out of 24 datasets.
2. **Limited intuition for theoretical results.** Although the theory is sound, the paper should do a better job providing intuition on *why* the decoupling leads to improved long-context performance (see Question 2).
3. **Lack of detail in key arguments.** Some claims and statements would be clearer with additional justification or elaboration (see Question 3).

**Questions:**

1. **Missing baseline.** Why does Table 4 not report the performance of the LongPO baseline?
2. **Clarification on Figure 4.** What exactly do “efficiency” and “runtime” refer to? It is unclear how SoLoPO improves computational efficiency, as it requires handling both $ x_{\text{short}} $ and $ x_{\text{long}} $, which requires additional time for dataset creation as well as an additional forward pass for every iteration. Could you clarify where the runtime or efficiency advantage arises?
3. **Clarification on Theorem 1 and the function $ s(\cdot) $.**
   1. Theorem 1 introduces $ s(|x|) $. It would be helpful to provide intuition for the role of $ s(\cdot) $ and its relationship with $ f $, beyond the theoretical requirement for this assumption.
   2. Since $ s(\cdot) $ must satisfy the inequality, multiple valid functions may exist. Would a tighter $ s $ lead to a stronger upper bound, and how would this affect empirical behavior?
   3. It would improve readability to include some intuition, after Theorem 1, on how $ s(\cdot) $ can be derived for a given alignment method. The appendix (I3, I4) provides examples, but referencing them directly in the main text would help readers.
4. **On the equivalence with standard PO.** Line 201 states that for $ \rho = 100\% $, SoLoPO is equivalent to the original PO. In this case, we have
   $$
   L_{\text{PO}}(x_{\text{long}}) \le \tfrac{1}{3} L_{\text{SoLoPO}}(x_{\text{short}}),
   $$
   which is an upper bound. Is there theoretical evidence that this bound becomes an equality? If not, have you empirically tested whether optimizing PO or its SoLoPO upper bound leads to equivalent results when $ x_{\text{short}} = x_{\text{long}} $?

**Minor comments that do not affect rating**
1. In Table 4 (Open LLM Leaderboard), best and second-best values are not bolded or underlined, unlike other tables. Any reason for this inconsistency?
2. While Assumption 1 appears reasonable, it would strengthen the work to provide empirical evidence that it holds in practice.
4. What is the base model used in the LongPO reimplementation?
5. In Figure 1a, the second box should read “short-context preference optimization.”
6. Line 159 defines $ x_{\text{short}} $ as concatenating $ c_{\text{rel}} $, but the equation shows $ c_{\text{irr}} $. This appears to be a typo.
7. Section 4.1 would be stronger if it explicitly stated how many scenarios SoLoPO outperforms the corresponding PO baseline, further reinforcing its general applicability.
8. In Table 4, the result for $ M_{\text{short}}^{\text{SimPO}} $ in the Math column is shown in red, even though it improves over the Instruct model. Please verify the formatting.

---

> ### Author Response · Authors · 2025-11-20
> **Response to Reviewer mA21**
>
> We sincerely thank the reviewer for the comprehensive assessment and for highlighting the strengths of our work, especially the recognition of the novel SoLoPO framework for transferring short-context capabilities to long-context scenarios. We are pleased that you acknowledged its solid theoretical foundation, general applicability to various PO methods, and the strong performance of the chosen-only variant across multiple long-context benchmarks. Based on your insightful feedback, we have provided the following responses:
>
> > **Weakness1: Regarding the degraded short-context performance of SoLoPO compared to PO baselines.**
>
> Thank you for your careful observation.
> - Related works [1,2] indicate that during long-text training, a model's short-context capabilities may degrade relative to its initial state (e.g. instruct models), potentially due to "catastrophic forgetting." Therefore, our claim that "SoLoPO does not compromise short-context capabilities" is primarily made in comparison to the instruct model, rather than other baselines. As observed in Table 4, only SoLo-DPO exhibits a slight decrease in average performance compared to Instruct and other DPO approaches. However, these drops are within acceptable ranges, as SoLoPO offers improvements in both the performance of long-context understanding and training efficiency. Additionally, even in cases of "catastrophic forgetting", it can be mitigated through simple data replay[1,2].
> - We will explicitly clarify this in the revised version to prevent any potential misunderstandings.
>
> > **Weakness2: Regarding a detailed intuition on why decoupling leads to improved long-context performance.**
>
> Thank you for the valuable feedback. We agree that providing stronger intuition for our theoretical results is important.
> - Our approach stems from the observation that long-context understanding tasks, such as multi-document QA, require two distinct capabilities:
>     - (1) contextual knowledge localization: identifying task-relevant knowledge from the context
>     - (2) contextual knowledge reasoning/utilization: effectively utilizing this knowledge to solve the problem.
> - Standard preference optimization algorithms primarily focus on modeling the relationship between outputs, thereby enhancing overall model performance. However, they do not explicitly encourage the model to learn the ability to locate crucial information within the context, which is inherently tied to the input.
> - To address this, we propose SoLoPO, a decoupled approach which explicitly models the relationships between inputs and outputs separately. As described in Line 207, this strategy is designed to jointly foster the model's two crucial capabilities for long-context scenarios. Furthermore, we provide a detailed discussion of this in Appendix I.7 (Line 1632).
>
> > **Weakness3&Question3: Regarding clarification on Theorem 1 and the function $s(x)$.**
>
> Thank you for your constructive feedback. Due to previous length constraints, we will incorporate your suggestions in the revised version to enhance the paper's readability.
>
> - **Regarding the rationale behind using $s$**:
>     - The design of the function $s$ originates primarily from the derivation process of Theorem 1. Our core intention was to **unify** the two terms, $r_\phi(x_{long}, y)$ and $r_\phi(x_{short}, y)$, into a single expression. This unified expression serves to **quantify the distance** between $r_\phi(x_{long}, y)$ and $r_\phi(x_{short}, y)$. Building upon this foundation, we further generalized this concept in Appendix I.5 (Line 1603), leading to a more generalized theorem. We believe this generalized theorem may provide valuable insights and inspire future research directions.
>
> - **Regarding the impact of the tightness of $s$ (theoretically and empirically) on performance**:
>     - Theoretically, a tighter $s$ corresponds to a stronger upper bound, which may improve empirical performance. In contrast, a loose upper bound often results in vague optimization directions and may cause the optimization to converge to a local optimum. For example, consider minimizing $x^4-x^2=x^2(x^2-2)$, whose global minima are at  $x= ±1$. If we use an upper bound $x^4$, the global minimum of this upper bound is at $x=0$. Therefore, if the upper bound is too loose, the obtained solution may deviate from the true optimum.
>     - **Empirical Validation**:
>         - Setting: We compare a *tighter* setting $s_t(x) = |x|$ (used in the paper) with a *looser* setting $s_l(x) = |x| + \sin(x) + 1 \ge s_t(x)$ on Qwen2.5-7B-Instruct, while keeping all other training configurations identical.

---

> ### Author Response · Authors · 2025-11-20
> **Response to Reviewer mA21**
>
> | Model | QAs-LongBenchV1 |  |  | QAs-RULER |  |  |  |  |
> |---|---|---|---|---|---|---|---|---|
> |  | S-Doc QA | M-Doc QA | **Avg.** | 4k | 8k | 16k | 32k | **Avg.**|
> | $M^{DPO}_{SoLo(s_l)}$ |35.3 | **57.8**| 46.5|64.2|62.6 |61.2 |57.4|61.4|
> | $M^{DPO}_{SoLo(s_t)}$ |**38.0**| 57.6 |**47.8** |**66.4**| **64.5** |**62.7** |**57.7** |**62.8** |
> ||
> | $M^{SimPO}_{SoLo(s_l)}$ |36.8 |56.7| 46.7|67.8|65.0 |61.9|57.2|63.0|
> | $M^{SimPO}_{SoLo(s_t)}$ | **38.1** |**58.6**| **48.4**| **69.2** |**66.0** |**62.7** |**57.8** |**63.9**|
> ||
> | $M^{ORPO}_{SoLo(s_l)}$ |36.1| 57.8|  46.9|68.1|65.4 |60.5 |56.2|62.6|
> | $M^{ORPO}_{SoLo(s_t)}$ |**37.6**| **61.4** |**49.5**| **70.8**|**68.3** |**64.0**| **57.3** |**65.1** |
> ||
> - Analysis: As shown in the table above, the tighter $s(\cdot)$ generally achieves better performance, except in the MD-QA scenario of LongBenchV1 for SoLo-DPO. Since $s_t \leq s_l$ in our experiments, we can infer that a looser $s(\cdot)$ tends to yield worse performance.
> - **Regarding the readability suggestion related to Theorem 1 and $s$**:
>     - We will incorporate your feedback in the revised version by referring to the examples in Appendix I3 and I4 to explain how the alignment strategy is derived from $s$.
>
> > **Question1: Missing LongPO baseline in Table 4.**
>
> Thank you for noticing this detail. Initially, our primary focus was on analyzing SoLoPO's generalization and stability across different context lengths, which is why the results for LongPO were not included. We have now supplemented the results for LongPO on LongbenchV2 and the OpenLLM Leaderboard, as presented below:
>
> ||LongBenchV2|||||| Open LLM Leaderboard||||||
> |---|---|---|---|---|---|---|---|---|---|---|---|---|
> | **Model** |**Overall**| Easy | Hard | <32k | 32k-128k | >128k | MMLU-Pro | IFEval | BBH | MATH | GPQA | **Avg.** |
> | Qwen2.5-7B-Instruct | 29.3(±0.7) | 30.9 | 28.3 | 36.9 | 24.6 | 26.1 | 44.63 | 74.22 | 55.25 | 36.86 | 31.88 | 48.56 |
> ||
> |LongPO(reimplemented)|29.6(±1.5)|32.2|28.0|36.7|26.7|23.7|44.80|73.86*|55.07*|34.81*|31.91|48.08*|
> |LongPO(Open-source)|*33.3*(±0.5)|35.0|*32.0*|**40.5**|*30.0*|27.8|44.69|76.49|53.94*|32.32*|31.87*|47.86*|
> ||
> | $M^{\text{DPO}}_{\text{SoLo}}$ |**35.2(±1.2)**|**37.5**| **33.8** |39.3 |**31.8** |**35.0** |44.66 |73.98* |54.78* |35.57* |31.63* |48.12*|
> | $M^{\text{ORPO}}_{\text{SoLo}}$ | 33.2(±1.0) | *36.3*|31.2 |*39.7*|28.8 |*30.9*| 44.83 | 75.18 | 55.23* | 37.16 | 31.46* | 48.77|
> | $M^{\text{SimPO}}_{\text{SoLo}}$ | 31.0(±1.3) |34.1 |29.1 |37.5 |25.7 |30.6 |44.78 |75.90 |54.89* |37.76 |31.54* |48.97|
> - Notes:
>     - The backbone model is  Qwen2.5-7B-Instruct;
>     - 'Reimplemented' refers to the results obtained by training LongPO on our dataset;
>     - LongPO is based on DPO;
>     - \* indicates performance lower than the Instruct model.
> - Analysis:
>     - **Better Length Generalization of SoLoPO**: On LongBenchV2, SoLo-DPO outperforms LongPO (open-source) even though the latter uses more and longer training data. This may be because LongPO (open-source) directly sets $y \sim \pi(y|x_{\text{long}})$ as $y_l$ and $y \sim \pi(y|x_{\text{short}})$ as $y_w$ without ensuring $y_w \succ y_l$. In contrast, our method uses ground-truth to construct preference data, ensuring correctness and quality. On the same dataset, LongPO (reimplemented) also performs worse than SoLo-DPO, further validating SoLoPO’s advantage in length generalization.
>     - **Better Length Stability**: On the Open LLM Leaderboard, DPO-based methods show slightly lower performance compared to instruct model, but remain within an acceptable range, indicating good length stability for SoLoPO.
>     - **Better Performance of Other Variants** — Both SoLo-ORPO and SoLo-SimPO exhibit better length generalization and stability compared to LongPO (reimplemented).
> - Thank you again. In the revised version, we will include the corresponding content along with a detailed analysis. The results presented above further support SoLoPO’s advantages in both length generalization and length stability.

---

> ### Author Response · Authors · 2025-11-20
> **Response to Reviewer mA21**
>
> > **Question2: Regarding clarification on Figure 4 and an explanation of how SoLoPO's efficiency advantage arises.**
>
> - The “runtime” in Figure 4 represents model training time.
>
> - **How SoLoPO's efficiency advantage arises**:
>     - **Data Construction**: SoLoPO currently employs two primary methods for dataset construction:
>         - (Method 1: Augmenting Short Texts to Long): This approach starts with existing short-text and question-answer pairs $(c_{short}, q, a)$. Irrelevant text segments are randomly appended to these short texts to create longer contexts, $c_{long}$. This yields data tuples of the form $(c_{long}, c_{short}, q, a)$, which is the method adopted in this paper.
>         - (Method 2: Chunking and Generating from Long Texts): This method begins with an existing long text, $c_{long}$, which is then divided into multiple chunks. A corresponding question-answer pair $(q, a)$ is generated for each chunk $c_{short\_i}$ using an LLM.
>
>         - Furthermore, we directly sample from the short text $c_{short}$ and the question $q$ to construct preference data pairs $(y_w, y_l)$. Compared to directly sampling preference data from long texts, sampling from short texts is computationally more efficient:
>             - Shorter lengths require less time to sample the same number of instances.
>             - Directly sampling from long texts may exceed the LLM's effective processing window, further increasing sampling difficulty.
>     - **Model Training**:
>         - As detailed in Section 2.3, when we practically apply SoLoPO, we utilize a *chosen-only* SoLoRA approach. This involves processing $x_{short}$ twice but $x_{long}$ only once per training step. In contrast, the original PO algorithm requires processing $x_{long}$ twice. Consequently, SoLoPO reduces the computational cost associated with processing long texts. Since short-text processing is typically less costly than long-text processing, SoLoPO can lower training time and GPU memory overhead.
>         - In Appendix H.5, we provide a brief theoretical analysis. When the compression ratio is less than 0.7, SoLoPO can offer efficiency gains. If the compression ratio is larger (i.e., the short text length approaches the long text length), the efficiency gains of SoLoPO diminish. However, performance gains may still exist, as SoLoPO explicitly models the two key capabilities of contextual knowledge localization and utilization.
>
> - Due to previous space constraints, we will provide a proper explanation in the main text of the revised version to enhance the readability of the paper. Thanks for your question.
>
> > **Question4: Regarding how SoLoPO becomes equivalent to standard PO when $x_{long}$=$x_{short}$.**
>
> - Thank you for your careful review. In Equation (7), when $x_{long}=x_{short}=x$, we have: $$L_{\eta,\gamma}(x)\le \frac 13L_{3\eta,\frac13\gamma}(x)$$. Expanding further, we obtain:$$\mathbb{E}[f(\eta(r_\phi(x,y_w)-r_\phi(x,y_l)-\gamma))]\le\frac13\mathbb{E}[f(3\eta(r_\phi(x,y_w)-r_\phi(x,y_l)-\frac13\gamma))]$$. Both of these expressions are formally consistent with the standard preference optimization objective. They are equivalent in form, differing only in hyperparameters $\eta$ and $\gamma$. Therefore, when $x_{long}=x_{short}$, the SoLoPO loss naturally degenerates to the original standard PO form. Due to previous length constraints, we will explicitly clarify this in the revised version to enhance the paper's clarity.

---

> ### Author Response · Authors · 2025-11-20
> **Response to Reviewer mA21**
>
> > **Minor comments on the clarity of this paper：**
> - **Regarding table 4: why are the maximum and second-highest values not marked?**
>     - The Open LLM Leaderboard primarily aims to evaluate whether different models exhibit performance drops on short-context tasks compared to their Instruct versions. Therefore, we mainly highlight the performance degradation relative to the Instruct model rather than marking the absolute maximum and second-highest values.
> - **Regarding empirical evidence for Hypothesis 1.**
>     - We provide the following additional empirical experiments to support Hypothesis 1, and these results will be included or referenced in the revised version:
>         > *since $x_{\text{long}}$ contains more task-irrelevant information than $x_{\text{short}}$, making it harder for the model to distinguish preferences when conditioned on $x_{\text{long}}$:*$$p(y_w\succ y_l | x_{long}) \leq p(y_w\succ y_l | x_{short})$$where $p(y_w \succ y_l \mid x) = \sigma\left(r^\*(x,y_w) - r^\*(x,y_l)\right)$, $r^\*$ denotes the reward model, and $\sigma$ is the sigmoid function.
>     - In the absence of an optimal reward model $r^*$, we employ a model $\pi_{final}$ trained via DPO to estimate the reward margin for $(y_w, y_l)$, following the reward computation formulation defined in DPO.
>     - Specifically, we adopt Qwen2.5-7B-Instruct as the reference policy $\pi_{\text{ref}}$, and perform DPO training on data with a context length of $1K$ to obtain the final policy $\pi_{\text{final}}$. We set the short-context length to $4K$, from which we sample preference pairs $y_w \succ y_l \sim \pi_{\text{ref}}(y \mid x_{\text{short}})$, and subsequently expand them to lengths ranging from $8K$ to $32K$ to obtain $x_{\text{long}}$. **This design aims to emulate a reward model with a non-trivial scoring capacity that is nonetheless susceptible to noise, in order to meet the preconditions of our assumption.**
>
>         | Length of $x_{long}$ | $8K$ | $12K$ |$16K$ |$20K$ |$24K$ |$28K$ |$32K$ |
>         |-------|-------|-------|-------|-------|-------|-------|-------|
>         |Hypothesis Validity Proportion|80.58%| 86.41% |91.26% |90.29%| 96.12% |95.15% |94.17%|
>     - The results are shown in the table above. As the context length increases, the amount of task-irrelevant information grows, and the proportion satisfying the hypothesis gradually rises, stabilizing at around 95%. Considering potential inaccuracies in reward estimation, such consistently high proportions provide substantial support for our Hypothesis.
> - **Regarding the base model used in LongPO experiments.**
>     - We use Qwen2.5-7B as the backbone, consistent with the LongPO work we follow and with the experiments in this paper, as described in Appendix E.1 MODEL TRAINING CONFIGURATION(line 1053).
> - **Regarding minor typos and the suggestion to clearly state in which scenarios SoLoPO performs better.**
>     - We sincerely thank you for the careful reading. We will correct the typos and also adopt your suggestion in the revised manuscript.
> ```
> [1] LongReD: Mitigating Short-Text Degradation of Long-Context Large Language Models via Restoration Distillation. (ACL 2025)
> [2] How to train long-context language models (effectively). (ACL 2025)
> ```

---

> > ### Author Response · Authors · 2025-11-27
> > **Request for Further Discussion**
> >
> > Dear Reviewer mA21,
> >
> > Thank you once again for the time and thoughtful effort you have put into reviewing our manuscript.
> >
> > We have carefully considered your valuable feedback, particularly regarding the clarity of our theoretical explanations, and have provided additional explanations and experiments in our response, along with corresponding revisions in the updated manuscript.
> >
> > As the discussion period is now in its final week, we would appreciate any remaining feedback on our response. We look forward to your further guidance and hope to address any remaining concerns before the discussion closes.
> >
> > Best regards,
> >
> > Authors

---

> > > ### Comment · Reviewer_mA21 · 2025-11-27
> > >
> > > I appreciate the authors' detailed replies and the extended experiment results. Since all of my concerns have been addressed, I have raised my score accordingly.

---

> ### Comment · Reviewer_mA21 · 2025-11-27
> **Rating update**
>
> Due to some technical issue, I am unable to edit my original review to update my rating.
>
> After carefully reviewing the authors' response, I would like to update my rating to "Accept (8)".

---

> ### Author Response · Authors · 2025-11-28
> **Appreciation for Your Constructive Review**
>
> Dear Reviewer mA21,
>
> Thank you for your thoughtful and constructive feedback on our work. We sincerely appreciate the time and effort you have dedicated to evaluating our work, as well as your careful consideration in deciding to raise your score.
>
> We are glad to know that our responses and revisions have adequately addressed your concerns. In accordance with your suggestions, we have made the following improvements in the revised version:
> 1. Added more detailed explanations of the theoretical derivations in the main text, along with references to illustrative examples in the appendix.
> 2. Provided further intuition on how SoLoPO enhances LLMs' long-context understanding and improves training efficiency.
> 3. Expanded the experimental section, including:
>     - Incorporating the LongPO baseline and a brief analysis in Table 4;
>     - Clarifying that SoLoPO's stability in short-context capabilities is relative to instruct models;
>     - Adding experimental verifications on Assumption 1 and the tightness of $s()$ in Appendices I.7 and I.8, respectively.
>
> Your feedback has been instrumental in helping us improve the quality and presentation of our work.
>
> Thank you once again for your diligent review and constructive guidance.
>
> Best regards,
>
> Authors

---

### Official Review · Reviewer_N1ah · 2025-11-01

**Soundness:** 3
**Presentation:** 4
**Contribution:** 3
**Rating:** 8
**Confidence:** 4

**Summary:**

This paper introduces SoLoPO (Short-to-Long Preference Optimization), a general framework for efficiently enhancing long-context reasoning in large language models. The key idea is to decouple long-context preference optimization (PO) into two stages: (1) short-context PO, which focuses on optimizing reasoning and alignment on short, information-dense contexts; and (2) short-to-long reward alignment (SoLo-RA), which aligns reward scores between paired short and long contexts that share essential task-relevant information.
The method can be plugged into existing PO algorithms such as DPO, SimPO, and ORPO. The authors derive a theoretical upper bound showing that the long-context PO objective can be approximated through this two-part decomposition, and propose a “chosen-only” SoLo-RA variant that further improves efficiency. Experiments on reasoning (LongBenchV1/V2, NIAH-Plus) and instruction-following benchmarks (MMLU-Pro, GPQA, BBH) demonstrate that SoLoPO achieves higher long-context performance, better reward–KL efficiency, and shorter training time without harming short-context capabilities.

**Strengths:**

1.The proposed decoupling framework (short-context PO + reward alignment) is elegant, theoretically grounded, and easy to integrate into existing RLHF/PO pipelines.

2.The theoretical formulation clearly explains how SoLoPO approximates the long-context objective through an upper bound, providing a solid foundation for the method.

3.The “chosen-only” SoLo-RA variant is an insightful practical contribution that reduces instability and significantly cuts training cost while maintaining effectiveness.

4.Extensive experiments cover multiple backbones (Qwen2.5-7B, Llama3.1-8B) and benchmarks, showing consistent gains in both long-context reasoning and efficiency.

5.The method generalizes well across DPO, SimPO, and ORPO, confirming its broad applicability.

6.The paper is well written, conceptually coherent, and supported by detailed ablations and efficiency analyses.

**Weaknesses:**

1.The paper could provide more qualitative analysis or visualization to show how SoLoPO improves long-context reasoning (e.g., attention heatmaps or retrieved key information patterns).

2.The framework assumes that short contexts can fully preserve essential information; performance may degrade if the summarization or compression is imperfect, which is not deeply discussed.

**Questions:**

None

---

> ### Author Response · Authors · 2025-11-20
> **Response to Reviewer N1ah**
>
> We sincerely thank the reviewer for the comprehensive and constructive review. We are grateful for your recognition of SoLoPO's theoretical foundation, elegant decoupling strategy and its high extensibility to different PO algorithms, as well as the significant efficiency gains of the chosen-only variant. Your comments on our comprehensive experiments and clear writing are also very encouraging. Based on your valuable feedback, we provide the following detailed responses:
> > **Weakness1: More qualitative analysis or visualization to demonstrate SoLoPO's enhancement of long-context reasoning.**
>
> Thank you for the valuable feedback.
> - In Appendix (Line 1261, Figure 9), we present heatmaps of model performance on NIAH-Plus when trained with different ORPO variants. Compared to directly using the Long-ORPO or Expand-Long ORPO method, SoLo-ORPO significantly improves the model’s ability to retrieve information across different depths and context lengths in both single-hop and multi-hop settings. This indicates that SoLoPO enhances the model’s capability to locate relevant information within long contexts, thereby improving reasoning in long-context scenarios. We will include a reference to this in the main text of the revised version to help readers better understand SoLoPO.
>
> > **Weakness2: Concerns about potential performance degradation due to imperfect compression methods.**
>
> Thank you for this insightful comment.
> - SoLoPO aligns with mainstream long-context data synthesis paradigms that ensure essential information is retained within short texts:
>     - Concatenation-based Synthesis: Starting from existing short-context data $(c_{\text{short}}, q, a)$, we concatenate multiple $c_{\text{short}}$ instances to form longer texts $c_{\text{long}}$[1,2].
>     - Chunking and QA Generation: From long-context data $c_{\text{long}}$, we split it into multiple chunks. Each chunk is treated as $c_{\text{short}}$ and used to generate corresponding QA or instruction-following data $(q,a)$[3,4].
>     - Both mainstream approaches ensure that the essential task information is fully preserved within the short-context data.
> - Since our current work mainly focuses on optimizing long-context training objectives and efficiency, we plan to conduct detailed experiments on the preservation of essential information in short-context data as part of future research on long-context data augmentation. We sincerely appreciate your valuable suggestion.
> ```
> [1] NExtLong: T oward Eﬀective Long-Context Training without Long Documents (ICML 2024)
> [2] Large Language Models Can Self-Improve in Long-context Reasoning
> [3] LongPO: Long Context Self-Evolution of Large Language Models through Short-to-Long Preference Optimization (ICLR 2025)
> [4] LOGO -- Long cOntext aliGnment via efficient preference Optimization
> ```

---

> > ### Author Response · Authors · 2025-12-03
> > **Appreciation for Your Constructive Review**
> >
> > Dear Reviewer N1ah,
> >
> > Thank you for the time and effort you invested in reviewing our work and for maintaining a positive attitude throughout the process.
> >
> > We have incorporated your suggestions in the revision.  Specifically, we now reference the comparative heatmaps (Figure 9, Appendix G) of ORPO variants' performace on NIAH-Plus in Section 4.2. This better demonstrates SoLoPO's enhanced contextual knowledge localization.
> >
> > We greatly appreciate your constructive feedback.
> >
> > Sincerely,
> >
> > Authors

---

### Author Response · Authors · 2025-11-21
**Updates in the manuscript**

We are very grateful for the valuable feedback from all reviewers. Based on these comments, we have revised our paper. All modified sections are highlighted in blue for easy comparison. The revisions can be summarized as follows:
> **Main Text:**
- **Clarifications on Theoretical Derivations**:
    - In Assumption 1, we added a reference to Appendix I.7 about the experimental evidence.
    - In Theorem 1, we have added the motivation for using the $s(\cdot)$ function and discussed the impact of its tightness.
    - In the section introducing SoLoPO's optimization objective, we have cited Appendix I3 and I4 regarding the derivations for $f(x)=x^2$ and $f(x)=\log \sigma(x)$ scenarios, along with Table 16, to enhance readability. Additionally, we clarified that SoLoPO and vanilla PO are formally equivalent when $\rho$=100\%.
- **Intuition on Decoupling for Long-Context Performance**: We have expanded on why decoupling improves long-context performance in the "What does SoLoPO learn?" (previously detailed in Appendix I9).
- **How SoLoPO  improves the Efficiency  of Long-context Alignment**: Section 2.3 now includes "How does SoLoPO improve data sampling and training efficiency?" to explain SoLoPO's benefits in data sampling and model training efficiency.
- **Experimental Section:**
    - Table 4 now includes the performance of LongPO, accompanied by a brief analysis.
    - We clarified that SoLoPO's stability in short-context capabilities is relative to instruct models.
    - We added a reference to Figure 9 in Appendix G, which visualizes different ORPO variants on the NIAH-Plus benchmark.
- **Discussion**: We added a reference to the discussion of limitations and future work in Appendix C, located after the conclusion.
- Revisions addressing Reviewer mA21's feedback on typos and readability improvements.
> **Appendix:**
- **Appendix C (Limitations & Future Work)**: Expanded more details on SoLoPO's potential in long-text generation tasks.
- **Appendix I.7**: Added experimental verification for Assumption 1.
- **Appendix I.8**: Added experimental verification for the tightness of $s(\cdot)$.

---

### Author Response · Authors · 2025-12-03
**Summary of Rebuttal Process**

We sincerely thank all reviewers for their time, thoughtful evaluations, and constructive feedback on our work.

We are grateful for all of the reviewers' positive feedback on SoLoPO, noting that:
- SoLoPO has a **solid theoretical foundation** and **the decoupling approach is novel and well-motivated**.
- Empirical evaluations across multiple backbone models (e.g., Qwen2.5, Llama3.1) and long-context benchmarks (e.g., RULER-QAs, LongBenchV2) demonstrate that SoLoPO with chosen-only SoLo-RA **enhances models' long-context understanding performance and markedly improves training efficiency**.
- SoLoPO is **easily extensible** and can be readily combined with mainstream PO methods such as DPO, SimPO and ORPO.

> **Summary of Responses to Reviewer N1ah** (**Rating: 8**)
   - **Regarding more visualizations for better readability**:
        - In the revised Section 4.2, we incorporated a reference to Figure 9 in Appendix F, which presents heatmaps of various ORPO algorithms on the NIAH-Plus.
   - **Regarding the impact of imperfect compression methods**:
        - We clarified that SoLoPO is compatible with mainstream long-context data construction methods (e.g., Concatenation-based Synthesis), all of which ensure that key information is fully preserved in the short contexts.

>  **Summary of Responses to Reviewer mA21** (**Rating: 4$\to$8**)
   - **Clarifications on Theorem 1**:
        - **Role of $s(\cdot)$**: We clarified that $s(\cdot)$ is introduced within the proof of Theorem 1 to quantify the distance between $r_\phi(x_{\text{long}}, y)$ and $r_\phi(x_{\text{short}}, y)$.
        - **Impact of $s(\cdot)$ tightness**: We provided a theoretical explanation that a tighter $s(\cdot)$ leads to better alignment and further added experiments confirming this effect.
        - **Equivalence with standard PO**: We explained that when $x_{\text{long}} = x_{\text{short}}$, SoLoPO becomes formally equivalent to standard PO, with differences pertaining solely to hyperparameters.
   - **Intuition Behind Decoupling's Benefit for Long-Context Performance:**
        - We explained that long-context scenarios necessitate models to possess two critical competencies: (i) contextual knowledge localization and (ii) contextual knowledge utilization. Standard PO methods, however, typically do not explicitly model the former, an ability intrinsically linked to the inputs. SoLoPO addresses this by explicitly decoupling the optimization problem, thus directly optimizing for both fundamental capabilities and more effectively aligning with the demands of long-context modeling. A detailed discussion is provided in Appendix I.9.
   - **Details on how SoLoPO improves efficiency of long-context alignment**:
        - We clarified that SoLoPO achieves this by sampling preference pairs based on short contexts and reducing long-context processing during training via chosen-only SoLo-RA.
   - **Missing LongPO baseline in Table 4**:
        - We integrated the results of LongPO into the revised Table 4, accompanied by a detailed analysis.
   - **Empirical verification of Assumption 1**:
     - We performed a supplementary experiment, which corroborates the assumption and is now documented in Appendix I.8.

Reviewer mA21 affirmed that all the concerns had been resolved, remarking: **"Since all of my concerns have been addressed"** and  further specifying: **"I would like to update my rating to 'Accept (8)'."**

> **Summary of Responses to Reviewer QMaK** (**Rating: 6**)
   - **On the Performance and Efficiency of SoLoPO in Fully Context-Relevant Scenarios**:
        - We explained that in cases where the long context is entirely task-relevant SoLoPO degenerates into standard PO.
        - We added experiments showing that models trained with SoLoPO on QA data exhibit no performance degradation on other fully context-relevant tasks, such as summarization. These results confirm SoLoPO's compatibility.
        - Appendix C (Limitations & Future Work) outlines discussions on exploiting the attention mechanism's inherent redundancy to achieve greater training efficiency in fully context-relevant tasks.
   - **Regarding SoLoPO's Extension to Long-Text Generation**:
        - We elaborated on the potential applicability of SoLoPO to long chain-of-thought , text refinement, and story generation tasks, with an expanded discussion provided in Appendix C.
   - **Regarding the concern about simple data construction lacking real-world complexity**:
        - We emphasize that SoLoPO demonstrates strong performance on representative real-world benchmarks, such as LongBench V1 and V2. This indicates the generalizability of this straightforward approach.
        - As further discussed in Appendix C, we defer the exploration of more sophisticated long-context data augmentation methods to future research.

   Reviewer QMaK confirmed the satisfaction with our responses, stating, "**I will maintain my positive score**."

---

### Meta-Review · Area_Chair_teLE · 2026-01-10

**Summary:**

This paper introduced a new preference optimization framework to enhance the long-context handling capability of LLMs. The core contribution is decoupling long-context preference optimization into short-context preference optimization and short-to-long reward alignment. The strength of this work is that the proposed idea is supported by both theoretical and empirical evidence, and also compatible with any preference optimization method (e.g., DPO, SimPO, ORPO). The extensive experimental results demonstrate the superiority of the proposed method.

**Reviewer Concerns:**

Initially, the reviewers raised some concerns on the experimental settings (e.g., missing baseline, more visualizations). Also, some reviewers requested clarification on the theorem and extension to long-text generation. But all of them are well resolved during the rebuttal periods, as explicitly denoted by the reviewers’ responses.

**Reviewer Scores:**

Initially, the reviewers' scores are (8,4,6). After successful rebuttal, the reviewer's scores are changed to (8,8,6). All reviewers are satisfied the responses and there are no remaining concerns that were raised by the reviewers, as explicitly stated by their responses.

---

### Decision · Program_Chairs · 2026-01-26

Accept (Poster)